# Mitochondria-specific photoactivation to monitor local sphingosine metabolism and function

Suihan Feng[1], Takeshi Harayama[1], Sylvie Montessuit[2], Fabrice PA David[3], Nicolas Winssinger[4], Jean-Claude Martinou[2], Howard Riezman[1]*

[1]Department of Biochemistry, University of Geneva, Geneva, Switzerland; [2]Department of Cell Biology, University of Geneva, Geneva, Switzerland; [3]Gene Expression Core Facility, Ecole Polytechnique Fédérale de Lausanne, Lausanne, Switzerland; [4]National Centre of Competence in Research (NCCR) in Chemical Biology, University of Geneva, Geneva, Switzerland

**Abstract** Photoactivation ('uncaging') is a powerful approach for releasing bioactive small-molecules in living cells. Current uncaging methods are limited by the random distribution of caged molecules within cells. We have developed a mitochondria-specific photoactivation method, which permitted us to release free sphingosine inside mitochondria and thereafter monitor local sphingosine metabolism by lipidomics. Our results indicate that sphingosine was quickly phosphorylated into sphingosine 1-phosphate (S1P) driven by sphingosine kinases. In time-course studies, the mitochondria-specific uncaged sphingosine demonstrated distinct metabolic patterns compared to globally-released sphingosine, and did not induce calcium spikes. Our data provide direct evidence that sphingolipid metabolism and signaling are highly dependent on the subcellular location and opens up new possibilities to study the effects of lipid localization on signaling and metabolic fate.

DOI: https://doi.org/10.7554/eLife.34555.001

*For correspondence:
howard.riezman@unige.ch

**Competing interests:** The authors declare that no competing interests exist.

## Introduction

Sphingolipids are one of the major lipid species in cellular membranes of all eukaryotic cells. Apart from maintaining structural properties of membranes, at least four sphingolipid metabolites, sphingosine, ceramide, sphingosine 1-phosphate (S1P) and ceramide 1-phosphate (C1P), are also signaling messengers that regulate fundamental cellular processes (*Aguilera-Romero et al., 2014*; *Hannun and Obeid, 2008*; *Maceyka and Spiegel, 2014*; *Atilla-Gokcumen et al., 2014*). Due to their essential roles, aberrant sphingolipid levels are linked to a broad range of diseases, including cancers, diabetes, inflammation and neurodegeneration (*Maceyka and Spiegel, 2014*; *Platt, 2014*; *Guri et al., 2017*). Interestingly, sphingolipid messengers could serve distinct functions depending on their subcellular localization, but this is currently difficult to address (*Hannun and Obeid, 2008*). For example, extracellular S1P activates receptors on the cell surface (*Lee et al., 1998*; *Van Brocklyn et al., 1998*; *Liu et al., 2000*), while S1P generated inside the nucleus has been proposed to regulate histone acetylation through direct interactions with the histone deacetylases (HDAC) (*Hait et al., 2009*). Furthermore, several protein domains have been suggested to bind to S1P, which might affect the recruitment of many PH domains (*Vonkova et al., 2015*). Although cellular localization is likely to be an important regulator of the metabolism and functions of sphingolipids, there is no direct evidence to demonstrate this.

In the last years some new techniques were introduced to explore lipid metabolism (*Wenk, 2005*; *da Silveira Dos Santos et al., 2014*; *Papan et al., 2014*) and to map lipid-protein interactions

**eLife digest** Fatty or oily molecules called lipids are essential components of the membranes of cells and important signaling molecules too. They are made in specific compartments of the cell, but most are found in all membranes, albeit in varying amounts. Their widespread distribution suggests that there are extensive networks for transporting lipids within cells. Yet scientists know little about lipid transport inside living cells because it is difficult to detect their movements.

Mitochondria are cellular compartments that are often referred to as the "powerhouses of the cell". Many lipids are found in mitochondria including one called sphingosine, which is a common component of many other cell membranes too. Sphingosine can increase the concentration of calcium ions inside the cells, and when converted to a molecule called sphingosine 1 phosphate it forms a signaling molecule that regulates fundamental processes like cell survival and migration. However, it was not known if sphingosine localized in the mitochondria was processed differently to the same molecule elsewhere in the cell, or if its signaling activity was affected by its location.

In the laboratory, Feng et al. synthesized an inactive sphingosine-like molecule that would only localize to mitochondria and which could be activated with a flash of light. By adding this molecule to human cells, they showed that sphingosine could be converted to sphingosine 1 phosphate within the mitochondria, before being exported rapidly to another compartment in the cell. The experiments allowed Feng et al. to observe the process in enough detail to to conclude that, despite its rapid transport, when localized only inside mitochondria, sphingosine could not trigger its normal signaling response.

This new light-activated lipid molecule will be a useful tool for many researchers studying both metabolism and signaling. In principle, a similar tool could be developed for many compounds and it should also be possible to localize the compound to different locations within the cell. This new generation of compounds would give scientists a better understanding of mitochondria biology. They could be applied to the study of diseases where the mitochondria do not function as they should, for example Barth syndrome, where a mitochondria specific lipid called cardiolipin is not properly synthesized.

DOI: https://doi.org/10.7554/eLife.34555.002

(*Haberkant et al., 2016*; *Saliba et al., 2016*). While proven to be valuable tools, they do not permit the acute manipulation of lipid amounts in living cells. Recently developed optogenetic (*Levskaya et al., 2009*) or chemical induced dimerization techniques (*Feng et al., 2014*) have been applied to rapidly change lipid levels by controlling the localization of lipid-metabolizing enzymes in living cells. Additionally, photoswitchable fatty acids and their derivatives also enables precise optical control of specific signaling lipid-protein interactions (*Frank et al., 2015*; *Frank et al., 2016*). However, successful design often requires detailed mechanistic understanding of the protein-lipid interactions. Alternatively, photoactivation ('uncaging') has become a common way to quickly increase the lipid supply. This method requires the chemical protection of the lipid head group using a photo-labile moiety, which blocks the lipid bioactivities. Once inside cells, a strong light flash can easily remove the protecting group within milli-seconds to seconds, and thereby release the bioactive lipids with high temporal resolution (*Höglinger et al., 2014*; *Klán et al., 2013*). Equipped with confocal fluorescence microscopy, the uncaging approach offers a convenient platform to study lipid signaling at the single cell level with minimal perturbation. To date, a number of caged lipids, including sphingosine (*Höglinger et al., 2015*; *Höglinger et al., 2017*), S1P (*Qiao et al., 1998*), fatty acids (*Nadler et al., 2015*), and phosphatidylinositol phosphates (PIPs) (*Subramanian et al., 2010*) have been successfully applied in lipid signaling studies.

To perform the photoactivation experiments, coumarin and nitrophenyl derivatives are widely used as caging groups to mask the bioactive small molecules (*Klán et al., 2013*). Nevertheless, the caged molecules are usually distributed in the cell randomly. Selective uncaging is only achieved through sophisticated fluorescence microscopy that is able to produce a highly focused laser beam in a defined area at the subcellular level. This method has disadvantages, though, because it is very difficult to exclusively activate the compartment of interest due to the diffraction limit of light. More importantly, this 'selective uncaging' strategy is limited to single-cell analysis, and hence is not

suitable for biochemical assays that are needed to study metabolism. Only a few exceptions allow uncaging with subcellular specificity, either from the outer leaflet of plasma membrane because the probes cannot enter into cells (*Nadler et al., 2015*), or in mitochondria to uncouple the membrane potential, but the latter system did not permit to visualize the caged compound once inside the cell making it difficult to determine its precise localization (*Chalmers et al., 2012*).

Here, we report a new mitochondria-targeted photoactivation method, which enabled the rapid release of sphingosine inside the mitochondria in living cells. We combined this method with lipid analysis by mass spectrometry (*da Silveira Dos Santos et al., 2014*; *Han, 2016*) to study mitochondrial sphingosine metabolism in intact cells. We observed that after photo-release, free sphingosine was rapidly phosphorylated into S1P, driven by sphingosine kinases. Inhibition of sphingosine kinases largely suppressed the S1P generation. We also applied this technique to monitor the turnover of sphingosine and compared the mito-caged sphingosine (Mito-So) with the globally distributed caged sphingosine (Sph-Cou) (*Höglinger et al., 2015*; *Höglinger et al., 2017*). In both cases, sphingosine was quickly lost indicating that these signaling sphingolipids are rapidly metabolized even when targeted to mitochondria. Using stable isotope-labeled caged sphingosine precursors, we investigated the conversion of sphingosine into ceramides and sphingomyelins after uncaging, and found that sphingosine metabolism was highly dependent on its subcellular localization. Mitochondria released sphingosine was used to continuously produce ceramides, whereas globally released sphingosine was more rapidly metabolized through sphingolipid metabolic network. Finally, we compared the two caged probes in calcium mobilization experiments. In contrast to the globally distributed Sph-Cou, the Mito-So failed to trigger calcium mobilization, demonstrating the importance of lipid localization in signal transduction.

## Results

### Synthesis and characterization of Mito-caged sphingolipids

Coumarin and nitrophenyl derivatives are the most popular caging molecules for biological studies. Although both can be efficiently cleaved from the caged molecules inside living cells, only coumarin-based molecules generate fluorescent signals and thus can be easily visualized under fluorescence microscopy. In order to determine the subcellular localization of caged molecules in cells, we chose to modify the 7-amino coumarin (**1**) by introducing a carboxylic linker and a methyl group. SeO2-mediated hydroxylation at the benzylic position afforded (**3**). TFA deprotection of the t-Bu ester allowed facile conjugation to a triphenylphosphonium (TPP) group via an amide bond to afford TPP-Cou-OH (**4**). Introduced by Murphy and co-workers (*Murphy, 1997*; *Murphy, 2008*), the lipophilic TPP cation is the best known mitochondria tag that facilitates accumulation of functional groups into mitochondrial matrix. Using the TPP-labeled coumarin alcohol (TPP-Cou-OH), we synthesized caged sphingosine (Mito-So) and sphinganine (Mito-Sa) following standard protocols (*Figure 1*).

To evaluate the photo-cleavage efficiency of the mitochondria-targeted caged lipids, we performed uncaging experiments in aqueous solution using a powerful UV lamp. Previously described diethylaminocoumarin caged sphingosine (Sph-Cou) was used as a reference compound (*Höglinger et al., 2015*). We analyzed the caging probes by LC-MS at different time points (*Figure 2B*). Most of the sphingosine was cleaved from the Mito-So after one minute, judging from the coumarin absorbance (*Figure 2B*). Compared to Sph-Cou, both Mito-So and Mito-Sa were completely cleaved after two minutes (*Figure 2C*), indicating the TPP-Cou-OH is an effective photolabile molecule. On the other hand, our measurement also indicated that the caged probes are fluorescent dyes that emit in the blue region (*Figure 1—figure supplement 1*). Next, we treated Hela cells with Mito-So for 15 min in the presence of MitoTracker. 5 µM of Mito-So was used to obtain sufficient signals since it is a relatively weak fluorophore. The pattern of Mito-So fluorescence completely overlaps with MitoTracker (*Figure 2D,E*, *Figure 2—figure supplement 1*), demonstrating Mito-So and Mito-Sa are predominantly targeted to mitochondria. Like other TPP-containing mitochondria targeting molecules (*Murphy, 2008*), this accumulation relies on the integrity of mitochondrial membrane potential. When the potential is disrupted by CCCP, the mitochondria staining of Mito-So was not seen (*Figure 2F*) and the probe was randomly distributed throughout the cell.

**Figure 1.** Synthesis of Mito-caged sphingosine (Mito-So) and sphinganine (Mito-Sa). Conditions: (i), tert-butyl bromoacetate, DIPEA, NaI, ACN, reflux, 80%; (ii) MeI, NaH, 0, 88%; (iii) SeO2, xylene, reflux; NaBH4, MeOH, 58%; (iv) TFA/DCM, 0; (3-Aminopropyl)triphenylphosphonium bromide, HBTU, DIPEA, DCM, 25%; (v) Bis-(4-nitrophenyl)carbonate, sphingosine or sphinganine, DIPEA, DMF, 60, 39–50%.

DOI: https://doi.org/10.7554/eLife.34555.003

The following figure supplement is available for figure 1:

**Figure supplement 1.** Photophysical properties of mitochondria-specific caged molecules.

DOI: https://doi.org/10.7554/eLife.34555.004

## Uncaged sphingosine and sphinganine are rapidly phosphorylated

In mammalian cells, sphingosine can be phosphorylated into pro-proliferative S1P, or converted into pro-apoptotic ceramide. To explore the metabolism of mitochondrial sphingosine, we performed lipidomics on Hela cells incubated with Mito-So and exposed to UV light (350–450 nm) for 2 min, which should be long enough to uncage most of the Mito-So (*Figure 2B*). Immediately after uncaging on ice, cells were collected for lipid extraction. The results show that both sphingosine and S1P levels were dramatically elevated compared to the controls (*Figure 3A,B*). Similarly, sphinganine and sphinganine 1-phosphate (Sa-1P) were also increased after uncaging of Mito-Sa (*Figure 3C,D*). Interestingly, the sphinganine level after Mito-Sa uncaging is higher compared to the sphingosine level after uncaging Mito-So; the opposite effects were observed in the phosphorylated form. Since we treated the cells with equal amount of the probes, the results might reflect the fact that sphingosine is better recognized as a substrate for phosphorylation and/or that that Mito-Sa is more efficiently accumulated inside the cells.

As a control, we examined the stability of Mito-So and Mito-Sa without photoactivation in the intracellular environments, and found no significant effects on sphingoid bases, showing that the Mito-caged lipids were sufficiently stable and were not cleaved by enzymatic activities such as esterases during the experiments. In addition, although sphingosine and sphinganine are structurally very similar, the sphinganine level remained stable after Mito-So uncaging (*Figure 3—figure supplement 1A*), and vice versa (*Figure 3—figure supplement 1B,C*). As a further control, we assessed the potential effects of UV light on the sphingolipid levels, by exposing the cells to continuous UV illumination without any caged probe and measuring the sphingosine and S1P levels. Although both lipids were slightly reduced after UV exposure (*Figure 3—figure supplement 2*), the reduction is rather minimal and is in the opposite direction compared to the effects of uncaging, showing that the

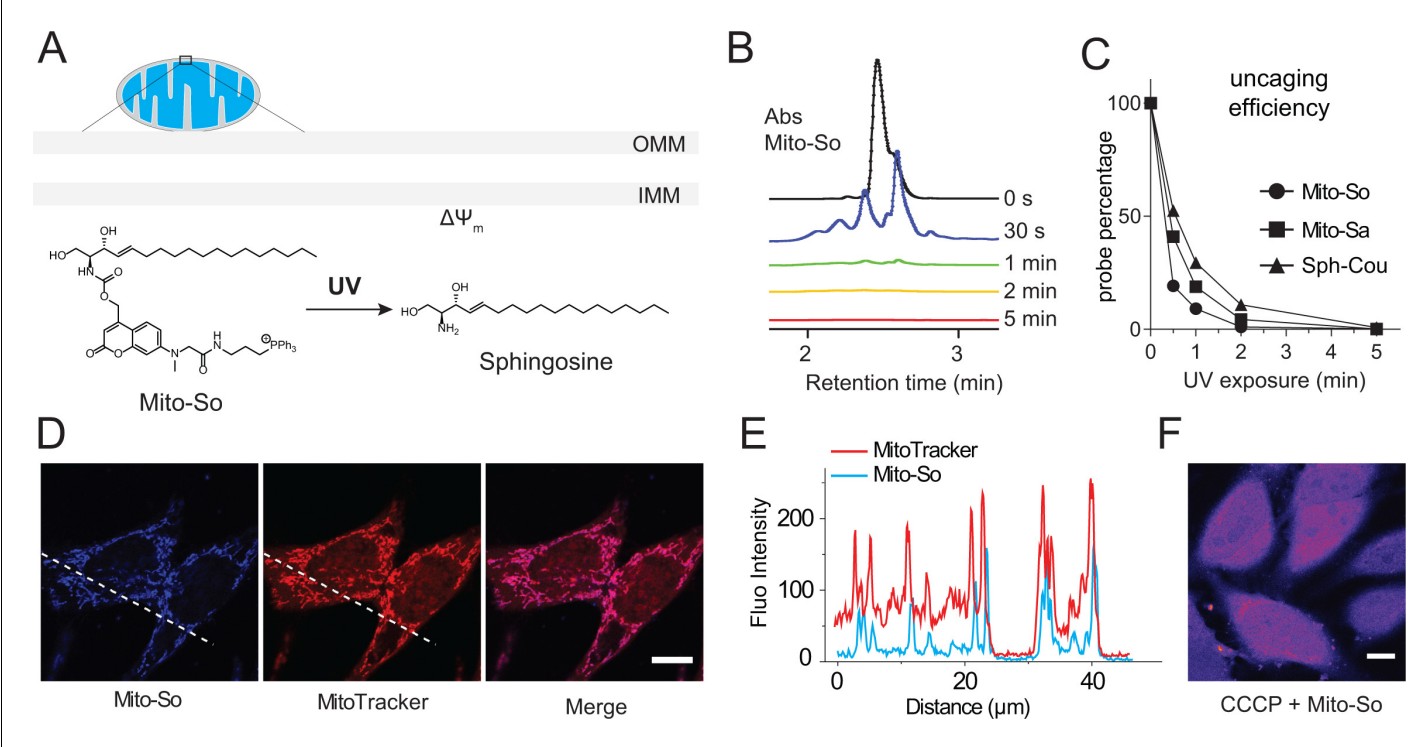

**Figure 2.** Uncaging efficiency and localization of mitochondria-targeted caged sphingosine. (A) Illustration of mitochondria-targeted photoactivation of sphingosine. (B) LC-MS spectral of Mito-So absorbance after exposing Mito-So (100 μM) solution (10% DMSO in water) under UV light for indicated time. (C) Uncaging efficiency of Mito-So, Mito-Sa, and Sph-Cou. All probes were treated using procedures described in (B) and quantified by the corresponding absorbance. (D) Representative fluorescence images of Hela cells stained with Mito-So (5 μM) and MitoTracker (50 nM). (E) Intensity profiles of the white dotted line (D) in Mito-So and MitoTracker channel, respectively. (F) Representative fluorescence images of Mito-So staining in the presence of CCCP (10 μM). Scale bar: 10.

DOI: https://doi.org/10.7554/eLife.34555.005

The following figure supplement is available for figure 2:

**Figure supplement 1.** Fluorescence images Mito-Sa staining in living cells.

DOI: https://doi.org/10.7554/eLife.34555.006

effect of UV illumination is insignificant under these conditions. Likewise, the major lipid species and ceramides were not affected after Mito-So and Mito-Sa uncaging (*Figure 3—figure supplement 1D,E*). Taken together, our results demonstrate that these photochemical probes, upon illumination, effectively released sphingoid bases with high spatiotemporal resolution. As sphingosine is the major sphingoid base in mammalian cells, we focused on Mito-So in the following studies.

Next, we sought to confirm that the elevated S1P after photo-releasing sphingosine required enzymatic phosphorylation and occurred after uncaging. Since sphingosine and S1P are interconvertible molecules, we assumed that the increased amount of S1P was due to phosphorylation of sphingosine by sphingosine kinases. There are two isoforms of sphingosine kinases in mammalian cells, sphingosine kinase 1 (SphK1) and sphingosine kinase 2 (SphK2), both of which are capable of phosphorylating sphingosine into S1P (*Maceyka et al., 2005*). To investigate the role of SphKs, we employed the CRISPR/Cas9-based genome editing strategy to generate a control and the double kinase knock-out cell line based on HeLa MZ cells(*Figure 4—figure supplement 1*) (*Ran et al., 2013*; *Liao et al., 2015*; *Harayama and Riezman, 2017*). After showing that removing sphingosine kinases did not disrupt mitochondria morphology or membrane potential (*Figure 4—figure supplement 2*), we carried out uncaging experiments using Mito-So in the cell lines and measured their sphingolipid levels. Consistent with previous results (*Figure 3*), sphingosine levels were dramatically increased after uncaging in the cells (*Figure 4A*). As expected, SphK double knock-out (dKO) cells failed to raise the S1P level after uncaging, in contrast to the control cells in which significantly elevated S1P was detected (*Figure 4B*). The level of S1P in the control cells was not as high as in

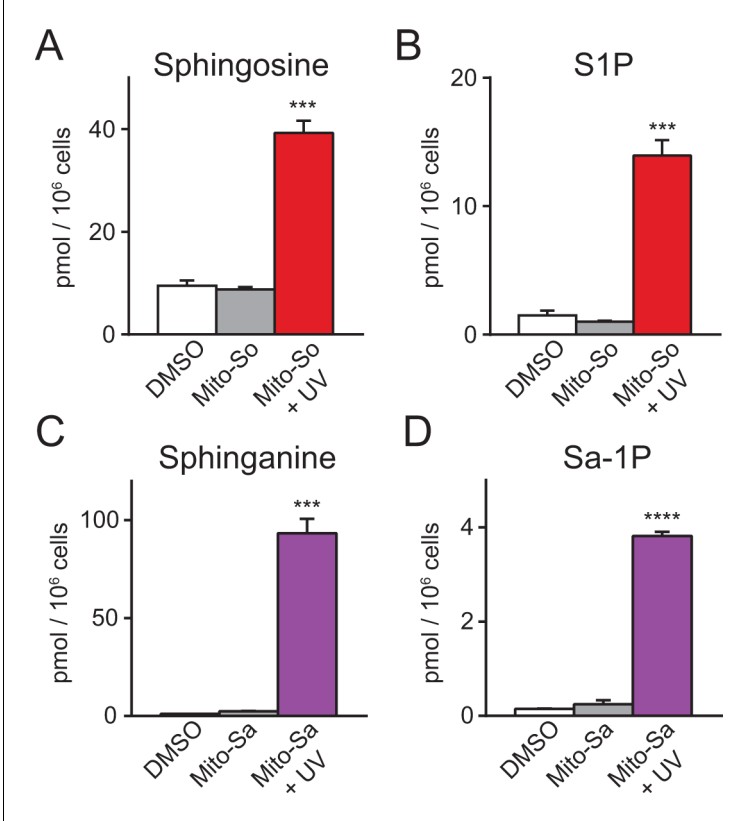

**Figure 3.** Sphingolipid analysis of mito-caged probes in Hela cells. Fully confluent cells in 60 mm culture dishes were incubated with the caged probes (2 µM) for 15 min, washed, irradiated under 350–450 nm UV light for 2 min on ice. Cells were collected immediately after UV irradiation. Lipids were extracted and measured by LC-MS. Values were normalized with respect to the amount of C17 internal standards and cell numbers. Data represents the average of three independent experiments. Error bars represent SEM.

DOI: https://doi.org/10.7554/eLife.34555.007

The following figure supplements are available for figure 3:

**Figure supplement 1.** Lipid analysis of mito-caged probes in Hela cells.
DOI: https://doi.org/10.7554/eLife.34555.008

**Figure supplement 2.** The effects of UV irradiation on sphingolipid levels.
DOI: https://doi.org/10.7554/eLife.34555.009

*Figure 3*, most likely due to the intrinsic differences between the cell lines used initially and the cell line used for generating knockout cells, even though at the beginning they were both selected from HeLa cells. Nevertheless, the Hela MZ is the most appropriate control for the mutant lines we produced.

To provide additional evidence supporting the role of SphKs in the S1P production, we performed uncaging experiments using HeLa cells in the presence of the sphingosine kinase inhibitor SKI-II, which inhibits both SphK1 and SphK2 (*French et al., 2003*). Consistent with the results in the knockout cells, SKI-II treated cells generated less S1P, and increased the level of sphingosine (*Figure 4C,D*). The SKI-II did not fully block S1P synthesis, which might be due to the fact that the SKI-II and sphingosine are competitive substrates for the kinases (*Lim et al., 2012*; *French et al., 2003*), and/or the access of SKI-II to the mitochondria might be limited. Taken together, our results clearly show that the rapid S1P accumulation after uncaging was catalyzed by sphingosine kinases.

As we detected a significant amount of S1P rapidly after uncaging, it is essential to know whether the S1P accumulation was driven by very fast enzymatic reactions, or if phosphorylation occurred on the caged sphingosine molecules (Mito-So) prior to UV illumination. Therefore, we extracted lipids from cells that were incubated with Mito-So, uncaged them after extraction and measured

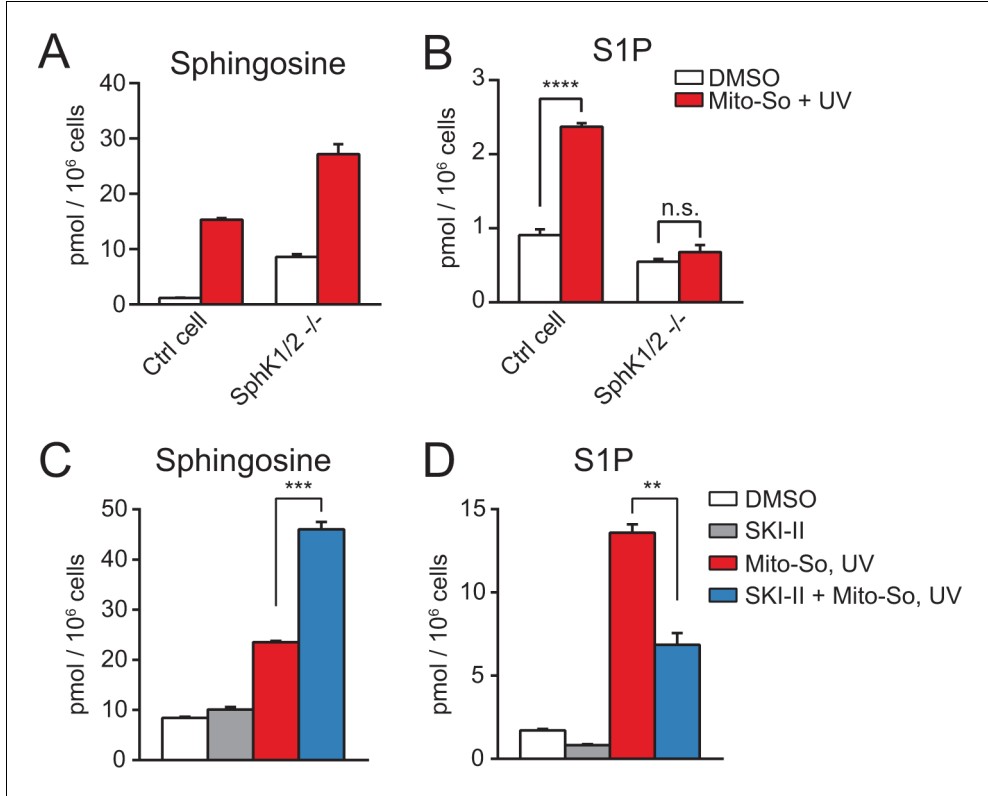

**Figure 4.** Sphingolipid analysis after Mito-So uncaging in sphingosine kinase double knockout Hela MZ cells (**A, B**) or inhibitor treated Hela cells (**C, D**). Values were normalized with respect to the amount of C17 internal standards and cell numbers. Data represents the average of three independent experiments. Error bars represent SEM. **p<0.01, ***p<0.001, ****p<0.0001, n.s., not significant, student's *t*-test.
DOI: https://doi.org/10.7554/eLife.34555.010

The following figure supplements are available for figure 4:

**Figure supplement 1.** Analysis of mutation rates in knockout cells.
DOI: https://doi.org/10.7554/eLife.34555.011

**Figure supplement 2.** Fluorescence images of Mito-So in SphK KO cells.
DOI: https://doi.org/10.7554/eLife.34555.012

**Figure supplement 3.** Mito-So was not phosphorylated in Hela cells in its caged form.
DOI: https://doi.org/10.7554/eLife.34555.013

sphingosine and S1P levels. There was no significant increase in S1P following this protocol, while, as expected, the amount of sphingosine was greatly increased (*Figure 4—figure supplement 3*). These results show that sphingosine kinases were not able to use the caged lipids as substrates in vivo even though the hydroxyl group was available, which further supports our conclusion that S1P accumulation is due to sphingosine kinase activity on the uncaged sphingoid bases.

Although the sphingoid bases were initially photo-released in mitochondria, we cannot formally rule out that free bases were rapidly transported out of the mitochondria and phosphorylated in the cytoplasm. To investigate this issue, we isolated mitochondria from mouse liver and performed uncaging experiments using Mito-So (*Figure 5*). Consistent with live-cell experiments, after UV irradiation, we detected an elevated sphingosine level (*Figure 5A*). Importantly, we also observed a significant increase in S1P, which was suppressed by the addition of the SKI-II inhibitor (*Figure 5B*). Compared with *Figure 3B*, the phosphorylation of sphingosine into S1P was less efficient in purified mitochondria, which might have been due to the different experimental settings and the intrinsic activity of purified mitochondria compared to mitochondria in vivo. For instance, it is possible that the SPHKs are more efficient in their native environments than in isolated mitochondria. In addition, if some of the S1P generated inside mitochondria was exported during the experiments it would

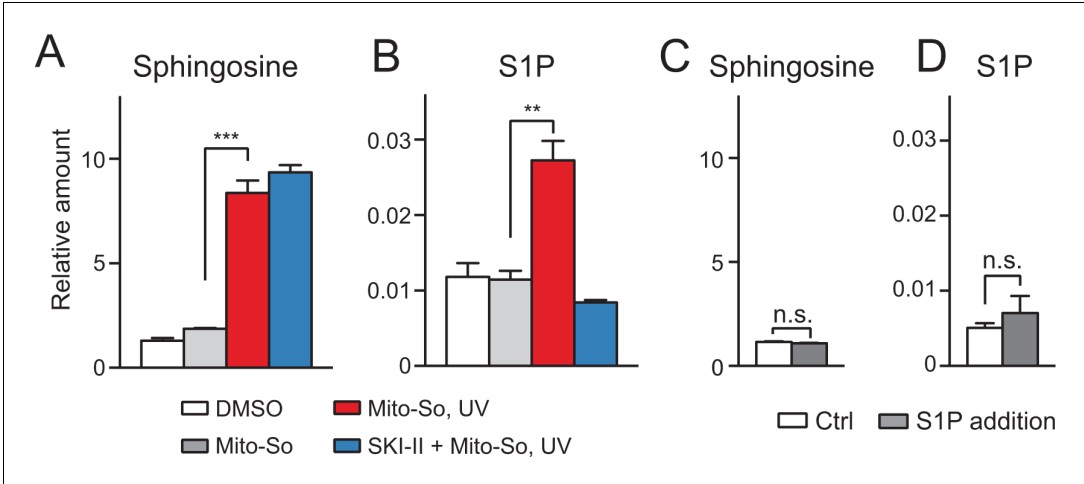

**Figure 5.** Sphingolipid analysis after Mito-So uncaging in purified mitochondria from mouse liver. (**A, B**) Mitochondria were incubated with Mito-So (10 µM) for 10 min, ±SKI II (50 µM), irradiated under UV light for 2 min on ice, incubated for 15 min at 37°C prior to pelleting by centrifugation and lipid extraction. (**C, D**) C18 sphingosine 1-phosphate (10 pmol, approx. 10 times of the control) was added to the mitochondria, following the same experimental and extraction procedures. Values were normalized with respect to the amount of C17 internal standards and cell numbers. Data represents the average of three independent experiments. Error bars represent SEM. **p<0.01, ***p<0.001, ****p<0.0001, n.s., not significant, student's *t*-test.
DOI: https://doi.org/10.7554/eLife.34555.014

have been lost during our isolation and extraction procedure (see following). To make sure the newly generated S1P was not caused by the contamination of mitochondria during purification and therefore produced outside of the mitochondria, we added a large amount of S1P (10 pmol, approx. 10 times more than produced in vitro) in a mock experiment without Mito-So to the purified mitochondria, and then proceeded with pelleting and lipid extraction, but did not detect any significant change of either sphingosine or S1P (*Figure 5C,D*). We therefore conclude that the S1P found in our assay was indeed inside mitochondria as extra-mitochondrial S1P was not detected in our assay, probably due to its relatively high solubility in aqueous buffers.

Collectively, our data demonstrate that, upon illumination, free sphingosine was quickly released from Mito-So in the mitochondria and was partially and rapidly converted into S1P by SphKs. The rapid phosphorylation, together with our results using purified mitochondria, provide convincing evidence that at least part of the released sphingosine was phosphorylated inside mitochondria. This is consistent with reports finding that sphingosine kinases are partially localized to mitochondria (*Lim et al., 2012*; *Strub et al., 2011*).

## Metabolism of locally uncaged sphingosine

Sphingolipids are key components of cellular membranes as well as signaling molecules involved in essential physiological and pathological processes (*Maceyka and Spiegel, 2014*; *Aguilera-Romero et al., 2014*; *Platt, 2014*). Their accumulation at specific intracellular locations could possibly be important for their physiological effects. Indeed, it has been proposed that sphingolipid signaling molecules could exhibit different cellular functions depending on their subcellular localization (*Hannun and Obeid, 2008*). Therefore, understanding their local metabolism is crucial to reveal the complexities of sphingolipid homeostasis and signaling.

To address the effect of subcellular localization on metabolism, we applied the photochemical probe to release sphingosine in mitochondria, and thereafter monitored the decay of sphingosine over time. We also performed parallel experiments using a recent reported caged probe (Sph-Cou, *Figure 6A* [*Höglinger et al., 2015*]) that releases sphingosine without subcellular selectivity. During the experiments, after 2 min uncaging on ice, the cells were incubated at 37°C and then were collected for lipid extraction at different time points (*Figure 6B*). Not surprisingly, in both cases, the released sphingosine was quickly metabolized without affecting the amounts of other major lipid

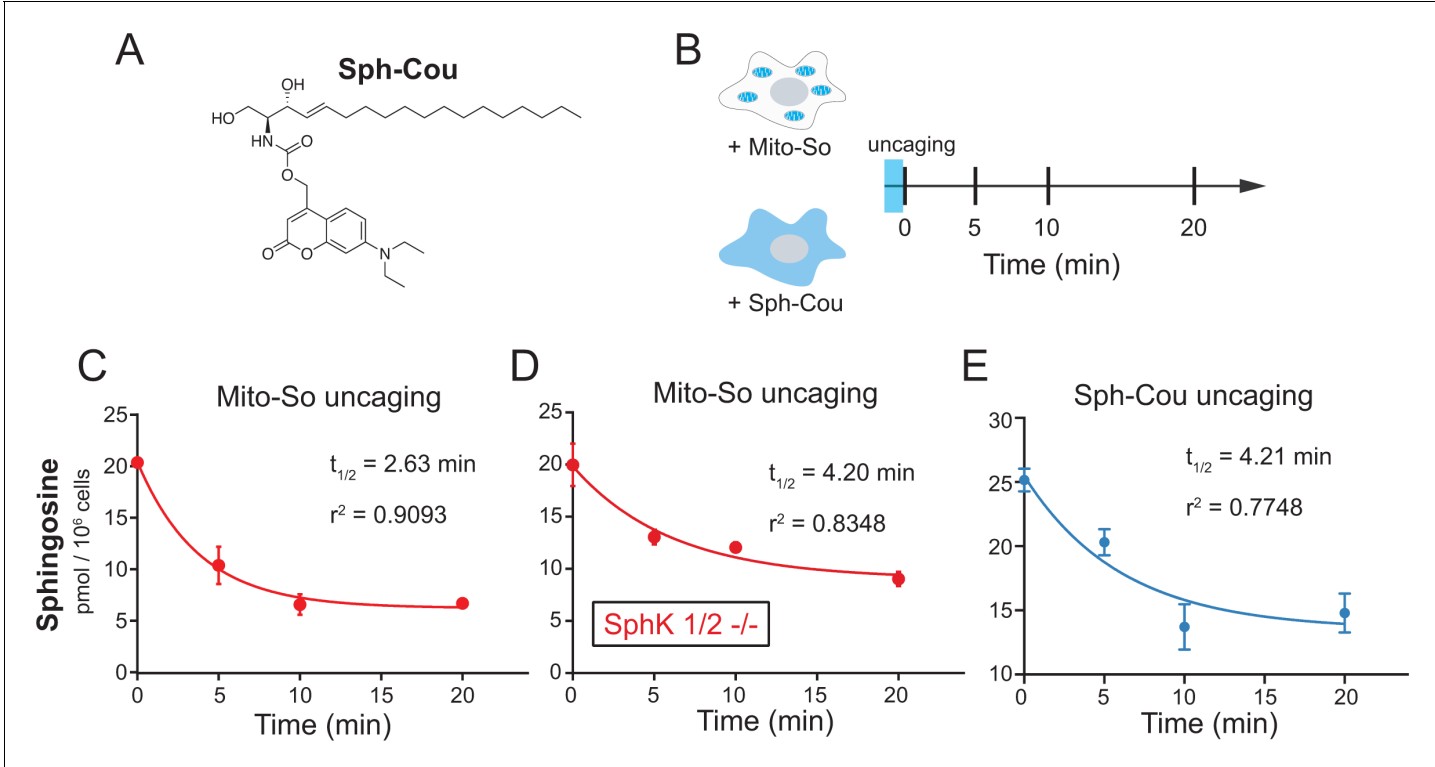

**Figure 6.** Real-time metabolism of mito-caged sphingosine and globally-caged sphingosine in living cells. (A) Chemical structure of Sph-Cou. (B) Illustration of uncaging followed by real-time detection. Cells were first treated with caged probes for 15 min, washed, irradiated by UV light for 2 min on ice, incubated at 37°C, and then collected for lipid extraction at different time points. (C, D , E) Changes of sphingosine levels over time after uncaging. The decay curves were generated by fitting into the one phase decay functions. Values were normalized with respect to the amount of C17 internal standards and cell numbers. Data represent the average of three independent experiments. Error bars represent SEM.

DOI: https://doi.org/10.7554/eLife.34555.015

The following figure supplement is available for figure 6:

**Figure supplement 1.** Total lipid analysis of mito-caged probes in Hela cells and SphK dKO cells.

DOI: https://doi.org/10.7554/eLife.34555.016

species (*Figure 6—figure supplement 1*). By simply fitting the time-course data with one-phase decay functions, we were able to estimate the half-lives of sphingosine. Notably, the decay of sphingosine ($t_{1/2}$ = 2.63 min, *Figure 6C*) generated from Mito-So is much faster than the one from Sph-Cou ($t_{1/2}$ = 4.21 min, *Figure 6E*). Also, unlike the Mito-So, sphingosine released from Sph-Cou did not drop to the basal level even after 20 min, suggesting that part of the globally released sphingosine was stored inside cells, which prevented further metabolism during the time course.

According to the sphingolipid metabolic network, sphingosine can either converted into ceramide by ceramide synthases, or be phosphorylated into S1P, which can be further degraded by sphingosine-1-phosphate lyase (SGPL), hydrolyzed to phosphate and sphingosine by lipid phosphatases, or secreted extracellularly (*Aguilera-Romero et al., 2014*). To explore the rate of conversion to ceramide, we uncaged Mito-So in SphK dKO cells, where the latter pathways are blocked due to a lack of formation of S1P and quantified the sphingosine level over time. Even though major lipid species were not affected (*Figure 6—figure supplement 1*), sphingosine levels still declined over time, but the half-life was significantly longer than in wild type cells ($t_{1/2}$ = 4.2 min, *Figure 6D*). Since S1P formation and sphingoid base turnover were blocked, the most logical explanation is that sphingosine turnover was the result of its conversion into ceramide by ceramide synthases, but this did not affect the total ceramide pool since steady state levels of ceramide in HeLa cells are much higher than sphingosine or S1P levels.

In order to further investigate the metabolic fate of sphingosine into more abundant lipid species, it is important to distinguish the newly formed ceramide from the endogenous ones. For this

purpose, we synthesized the d7-Mito-So and d7-Sph-Cou using heavy isotope labeled d7-sphingo-sine as a precursor. Following the same time frame and experimental settings, we measured the d7-sphingolipids over time after uncaging in live Hela cells (*Figure 7*). To better understand the sphin-golipid metabolism under various subcellular contexts, we performed experiments using direct addi-tion of the d7-sphingosine to cells, and compared this with the mito-released and globally-released sphingosine probes. Among the downstream sphingolipids, we focused on the formation of isotope-labeled C16 and C24 ceramides, the most abundant ceramides in HeLa cells, which also showed the most pronounced increase with negligible background signals in our study. As shown in *Figure 7* (left), we detected both d7-labeled ceramides immediately after uncaging of d7-Mito-So, demon-strating a rapid conversion to ceramide. The amount of newly formed ceramides steadily increased even after 20 min. Interestingly, globally localized d7-Sph-Cou generated a distinct metabolic pat-tern (*Figure 7*, middle). Fewer ceramides were formed at 0 min and both ceramides reached their maximum after 5 to 10 min and dropped significantly after 20 min, suggesting that the ceramides were quickly consumed to synthesize other sphingolipids. As for directly added d7-sphingosine, we observed continuous accumulation and sharp increase of ceramides over time, indicating that sphin-gosine entered from outside was efficiently metabolized by cells. As the most abundant sphingoli-pids on the plasma membrane, sphingomyelins are produced by sphingomyelin synthases using ceramides as precursors in the biosynthetic pathway. We also analyzed the signals of d7 C16 sphin-gomyelin and observed that, while the signals were increased over time in all cases, the signals gen-erated after global release of d7 Sph-Cou uncaging was close to reaching a plateau, unlike the externally added and mitochondrial released d7 Sph that were still in an early phase. The lower sphingomyelin signals, compared to ceramides, suggests that only part of the ceramide population was readily converted to sphingomyelin.

Sphingolipid metabolism is a complex process that involves a number of proteins and enzymes, and we are still in an early stage of our understanding. It has been suggested that sphingolipids serve distinct functions depending on their subcellular localizations, but it is technically difficult to address this issue. Our data provide direct evidence that subcellular localization is important for sphingolipid metabolism. The rapid turnover also suggests that lipid metabolism studies require techniques with high temporal resolution.

## Mitochondria-specific activation of sphingosine does not induce calcium mobilization

Recently, Höglinger *et al.* reported that uncaging sphingosine from Sph-Cou causes an acute release of calcium from acidic stores via the two-pore channel 1 (*Höglinger et al., 2015*). To investigate whether subcellular localization of sphingosine is relevant for this signaling event, we tested the Sph-Cou and Mito-So probes in live Hela cells using a ratiometric calcium dye (Fluo-4) as readout. As shown here, global uncaging of sphingosine quickly induced calcium release as previously reported, whereas mitochondria specific uncaging failed to trigger any calcium mobilization in the saame time frame ( *Figure 8*, *Figure 8—figure supplement 1*). To verify that this is not due to a difference in total cellular amounts of sphingosine, we quantified the sphingosine levels generated from the two probes extracted from cells. Since the calcium curves were obtained from single-cell analysis which does not provide quantitative information of photo-released sphingosine, we incubated the two probes in culture dishes, extracted lipids, performed uncaging in the lipid suspension, and measured sphingosine levels by mass spectrometry (same protocol as in *Figure 4—figure supplement 3*). The amount of sphingosine found after uncaging was approximately two times higher for Mito-So than for Sph-Cou (*Figure 8—figure supplement 2*), showing that differences in the amount of probe taken up by the cells is not the explanation for the different physiological consequences. Our data thus provide direct evidence that the intracellular sphingoid base compartmentalization can be a deciding factor in the regulation of intracellular signal transduction.

## Discussion

It is clear that glycerophospholipids and sphingolipids are unequally distributed among the subcellu-lar structures of the cell, however, the transport of lipids is very rapid, with phosphatidylethanol-amine being transported from its site of the synthesis to the plasma membrane in less than a couple of minutes (*Sleight and Pagano, 1983*). This rapid movement raises the question of the influence of

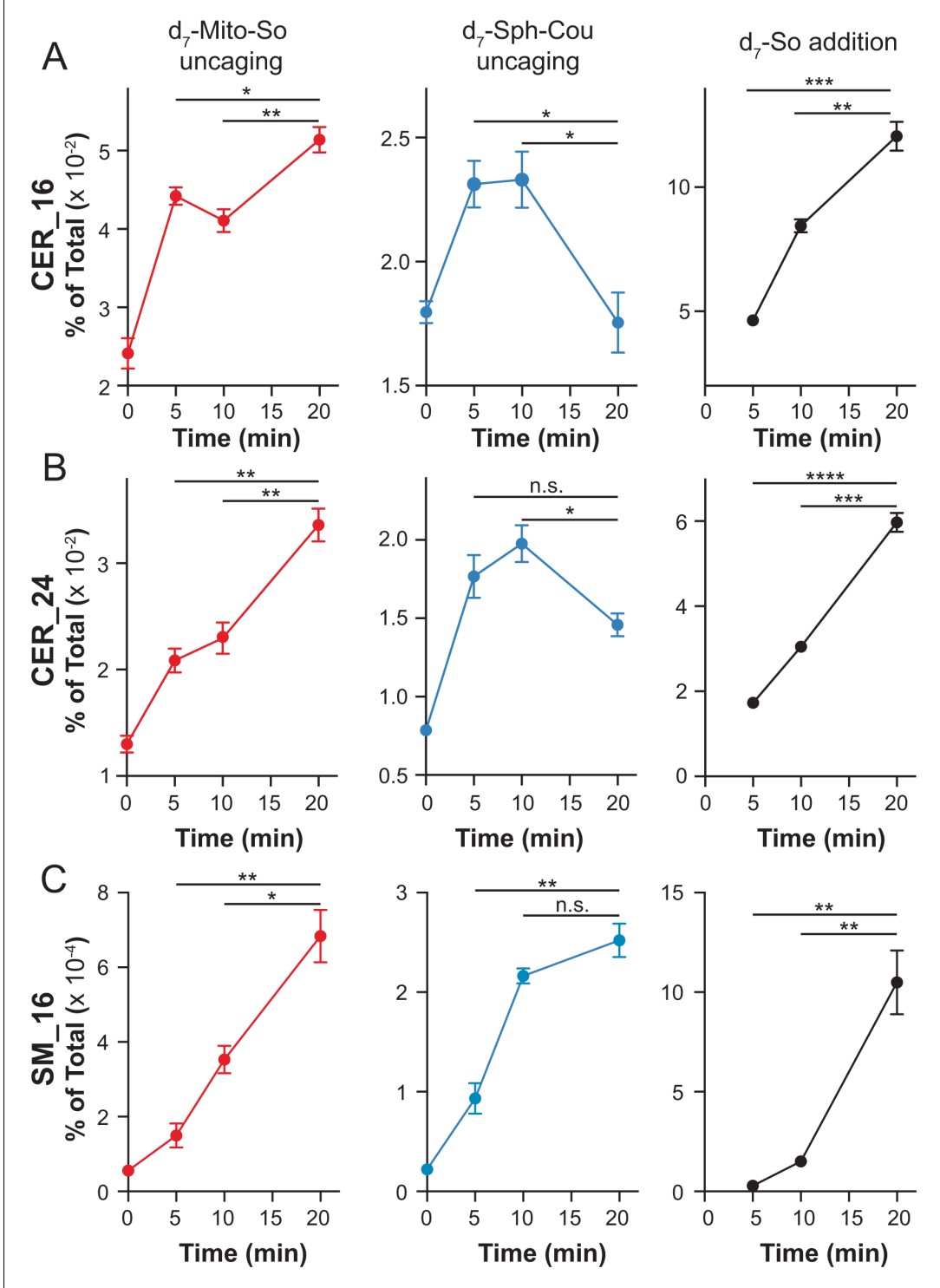

**Figure 7.** Real-time detection of d7-sphingolipids in living Hela cells after uncaging of d7-Mito-So, d7-Sph-Cou or direct addition of d7-sphingosine. Values were normalized with respect to the amount of internal standards and were plotted as percentage of all lipid signals. Data represent the average of three independent experiments. Error bars represent SEM. *p<0.05, **p<0.01, ***p<0.001, ****p<0.0001, n.s., not significant, students' *t*-test.

DOI: https://doi.org/10.7554/eLife.34555.017

The following figure supplement is available for figure 7:

**Figure supplement 1.** Total lipid profiles after releasing sphingosine in different subcellular compartments.

*Figure 7 continued on next page*

*Figure 7 continued*

DOI: https://doi.org/10.7554/eLife.34555.018

localization on metabolism and signaling. To address this, we have developed a highly efficient photo-cleavable probe (TPP-Cou-OH) that was easily installed on the amino group of free sphingoid bases, and which can be potentially applied to mask other functional groups, allowing the caging of a wide variety of lipids and other bioactive molecules. Carrying a hydrophobic and positively charged triphenylphosphonium (TPP) cation, the caged probes accumulated in mitochondria, into which the entry is tightly regulated. As these probes are also fluorescent dyes, their subcellular localization could be conveniently visualized under fluorescence microscopy. With few exceptions (*Nadler et al., 2015*; *Chalmers et al., 2012*), most available caged molecules are randomly distributed, and thus spatiotemporal control has to rely on sophisticated microscopy at a single-cell level. In contrast, our caged probes can be used for single-cell microscopic analysis as well as biochemical experiments to follow metabolism. In the recent years, mass-spectrometry (MS)-based lipidomic approaches have emerged as powerful tools that provide comprehensive and quantitative analysis of cellular lipid profiles (*da Silveira Dos Santos et al., 2014*; *Han, 2016*). Combining this approach with our caged probes produced a robust system that enabled us to monitor the local sphingosine metabolism in living cells.

We observed that, upon uncaging, some liberated sphingosine was rapidly converted into S1P. Suppressing sphingosine kinase activity, either by genetic ablation or chemical inhibitors, largely abolished S1P synthesis, confirming that phosphorylation was driven by sphingosine kinases. Importantly, uncaging sphingosine in purified mitochondria also generated a significant amount of S1P, providing strong evidence that mito-released sphingosine was phosphorylated inside mitochondria, consistent with previous studies that claimed SPHKs are detected in the mitochondria (*Strub et al., 2011*).

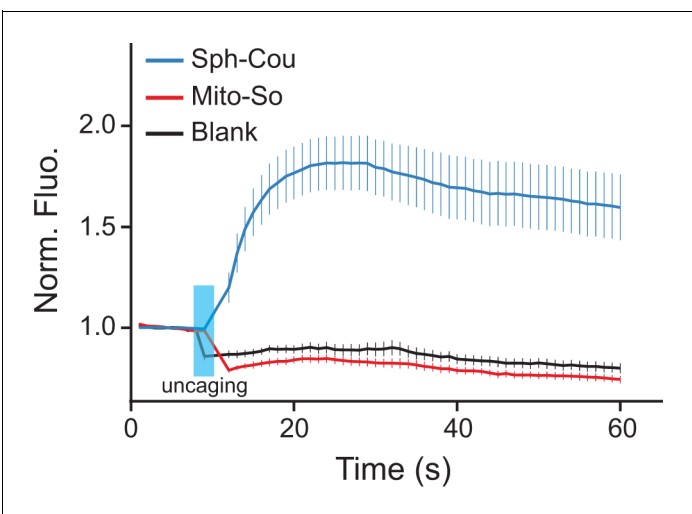

**Figure 8.** Calcium responses after photo-releasing sphingosine from caged precursors. (**A**) Mean traces of normalized fluorescence intensity after uncaging of Mito-So, Sph-Cou, or blank. Hela cells were loaded with Fluo-4 AM (5 µM), together with Sph-Cou (5 µM) or Mito-So (5 µM) prior to UV illumination. Cells were irradiated for 4 s by a 405 nm laser at 37°C. Error bars represent SEM. n > 10.
DOI: https://doi.org/10.7554/eLife.34555.019

The following figure supplements are available for figure 8:

**Figure supplement 1.** Histogram distribution of maximal calcium responses compared to the baseline in each cell, with the threshold set at 20% increase (black vertical line).
DOI: https://doi.org/10.7554/eLife.34555.020

**Figure supplement 2.** Comparison of cellular uptake between Sph-Cou and Mito-So.
DOI: https://doi.org/10.7554/eLife.34555.021

It has been known that ceramide is pro-apoptotic, whereas S1P promotes cell growth and proliferation. But the functions of sphingosine, which can be metabolized into ceramide and S1P, are not well understood. Since mitochondria is major a hub of energy supply and has been highly associated with apoptosis, it is interesting to investigate the metabolism of mitochondrial sphingosine, and to compare this with the metabolism globally distributed sphingosine. Using the mitochondria-specific caged probe (Mito-So) and globally caged probe (Sph-Cou), we monitored the sphingosine level over time, from which we derived the half-lives of decay by fitting a one phase function. Our results indicate that mitochondria released sphingosine was quickly lost over time ($t_{1/2}$ = 2.63 min, *Figure 6C*), whereas globally released sphingosine was slowly lost and did not come back to the basal level ($t_{1/2}$ = 4.21 min, *Figure 6E*).

Since ceramides are much more abundant lipids than sphingoid bases in cells ((*Simanshu et al., 2013*; *Shaner et al., 2009*; *Canals et al., 2010*), *Figure 3—figure supplement 1*), we prepared stable isotope labelled caged sphingosine so that the newly produced ceramides can be distinguished from the endogenous pool of ceramides by mass spectrometry. After releasing sphingosine at different subcellular localizations, we have found that sphingosine can be metabolized through distinct metabolic patterns, depending on where free sphingosine is supplied. Focusing on two most abundant ceramides in mammalian cells, our data show that both C16 and C24 ceramides were continuously generated after Mito-So uncaging. Nevertheless, after Sph-Cou uncaging, both lipids were produced and also quickly consumed after 5 to 10 min. Since ceramide synthases (CerS), which catalyze the conversion of sphingosine into ceramide, are mainly located in endoplasmic reticulum (ER), the lack of substrate accessibility to CerS may prevent immediate ceramide formation from liberated sphingosine in mitochondria. It is likely that the slow sphingosine exit from mitochondria resulted in the continuous increase of ceramides. On the other hand, globally uncaged sphingosine is easily accessible to CerS, which could explain the rapid ceramide production and degradation. We did not detect a preferential synthesis of ceramides with specific acyl chain length dependent upon the localized source of sphingosine. Interestingly, we detected less ceramides from globally released sphingosine. One possibility is that mitochondrial sphingosine is more efficiently, but less rapidly transferred to the ER, (*Figure 8—figure supplement 2*) the site of ceramide production. It is clearly shown from our data that sphingosine produced in the mitochondria can enter the sphingolipid biosynthetic pathway, demonstrating that there is a pathway to transfer sphingosine from the mitochondrial matrix to the ER. Overall, the data provided here provide direct evidence that sphingosine metabolism is different depending upon its subcellular localization.

To provide further evidence that sphingosine localization is important for signaling processes, we investigated the impact of mitochondria-specific uncaging in lipid signaling studies. We demonstrated a role of compartmentalization in restricting sphingosine signaling for calcium release. Again, sphingosine produced in the mitochondria was clearly not equivalent to sphingosine produced elsewhere in the cell. This clearly shows that the mitochondrial sphingosine does not freely and rapidly diffuse in the cells because it would have produced a release of calcium. Why does it not produce a calcium release? Two factors might influence this. One, it could be that the exit from mitochondria is too slow to be seen in our time course. Second, it could be that the mitochondrial sphingosine is transported preferentially to the ER. This would be consistent with the increased efficiency of ceramide synthesis from mitochondrial sphingosine compared to global sphingosine. It is possible that externally added sphingosine is also targeted preferentially to the ER as it behaves similarly to mitochondrial sphingosine. One interesting question for the future is to determine where the sphingosine must be located to induce a calcium release from acidic stores. A similar approach as taken here that localizes caged-sphingosine to different intracellular locations could be used to address this question and determine if the sphingosine must be inside or outside of the lysosome. Furthermore, extending our concept of subcellular release of caged lipids to other compartments and other lipids should allow experimentation to understand the role of localization in lipid metabolism and signaling in general.

Lipid messengers play active roles in essential cellular functions and their metabolism is tightly regulated through a complex lipid metabolic network. Although it is believed that sphingolipids could have different functions depending on their subcellular locations, it is difficult to acquire direct evidence because of technical challenges to spatiotemporally control bioactive lipids. Our approach offers a novel technique that permits the elucidation of the relevance of lipid localization in bioactive lipid metabolism and signaling. Combining this technique with mass spectrometry (MS)-based

lipidomic approach we were able to monitor real-time metabolism of a key lipid regulator in mitochondria. More importantly, our combined results provide direct evidence that sphingosine has distinct metabolism and signaling patterns depending on subcellular localization. We envision that site-specific photochemical probes will become valuable tools in lipid signaling and metabolism studies.

## Materials and methods

### Chemicals and reagents

Unless otherwise stated, chemicals and reagents were purchased from commercial sources (Sigma-Aldrich, Acros, Alfa-Aesar) and were used without purification. Sphingosine was purchased from Echelon Bioscience. Sphingosine-d7 was purchased from Avanti Polar Lipids. MitoTracker Orange CMTMRos was a gift from Jean Gruenberg lab (Department of Biochemistry, University of Geneva) originally purchased from Thermo Fisher Scientific. Fluo-4 AM was purchased from Thermo Fisher Scientific. Sph-Cou was synthesized according to previously described protocols (*Höglinger et al., 2015*). Deuterated solvents were obtained from Cambridge Isotope Laboratories, Inc. The plasmid pX330 was deposited to Addgene (plasmid #42230) by Feng Zhang (Broad Institute).

### Cell culture

All cells were cultured at 37°C and 5% CO2 in Dulbecco's Modified Eagle Medium (DMEM, Invitrogen) with 4.5 g/L glucose, supplemented with 10% fetal calf serum (FCS, Hyclone) and 1% Pen/Strep (Gibco). Cell numbers were quantified by Countess II Automated Cell Counter (Invitrogen) following manufacturer's protocol. For fluorescence microscopy experiments, cells were cultured in 35 mm glass bottom MatTek dishes to reach $_{80\%}$ confluency. In the subcellular localization experiments, cells were incubated with 5 µM Mito-So or Mito-Sa, together with 50 MitoTracker Orange in 1 mL imaging buffer (20 mM HEPES, 115 mM NaCl, 1.2 mM $MgCl_2$, 1.2 mM K2HPO4, 1.8 mM $CaCl_2$ and 0.2% glucose, pH 7.40) for 15 min at 37°C before replacing with new imaging buffer. In the calcium mobilization experiments, cells were incubated with blank, 5 µM Mito-So, or Sph-Cou, respectively, in the presence of 5 µM Fluo-4 AM in 1 mL imaging buffer for 15 min at 37 before replacing with new imaging buffer.

### Isolation and purification of mitochondria

Mouse livers were dissected, rinsed with ice-cold MB buffer (Mannitol 210 mM, Sucrose 70 mM, HEPES 10 mM, EDTA 1 mM, pH 7.5) and homogenized in 10 ml of MB buffer by twenty passages in a glass homogenizer. The resultant homogenate (liver homogenate) was centrifuged at 1500 g for 5 min at 4°C to remove nuclei and unbroken cells. The supernatant was further centrifuged twice at 6000 g for 10 min to provide a mitochondria-enriched pellet.

### Ethics statement

Mice were euthanized by $CO_2$ inhalation. All experimental procedures were performed according to guidelines provided by the Animal Welfare Act and Animal welfare ordinance, the Rectors' Conference of the Swiss Universities (CRUS) policy and the Swiss Academy of Medical Sciences/Swiss Academy of Sciences' Ethical Principles and Guidelines for Experiments on Animals, and were approved by the Geneva Cantonal Veterinary Authority (authorization number: 28038/GE86/16).

### Uncaging assays

Experiments were performed using a 1000 Watt Arc Lamp Source (#66924, NewPort) equipped with a dichromic mirror (350–450 nm, #66226). For live-cell uncaging, cells were seeded and cultured in 60 mm dishes until full confluency. The dishes were loaded with caged probes in 2 mL imaging buffer for 15 min at 37°C. After replacing with new imaging buffer, cells were placed on ice under the lamp at a distance of 20 cm, and irradiated 120 s at 1000 Watt. Cells were either placed back in a 37°C incubator, and/or immediately processed for lipid extraction. For the uncaging experiments in purified mitochondria, mitochondria in MB buffer was re-suspended in KCl buffer (125 mM KCl, 0.5 mM EGTA, 4 mM $MgCl_2$, 5 mM $K_2HPO_4$, 10 mM HEPES, pH 7.4) containing 5 mM ATP. Mito-So was added and the suspension was incubated at 37°C for 10 min, irradiated on ice for 2 min, re-incubated at 37°C for 15 min, centrifuged at 14,000 rpm for 5 min prior to lipid extraction.

## Lipid extraction and quantification

Lipids were extracted following previously described protocols with minor modifications (*Guan et al., 2009*; *Loizides-Mangold et al., 2012*; *da Silveira Dos Santos et al., 2014*). Briefly, cells were washed with cold PBS and scraped off in 500 µl cold PBS on ice. The suspension was transferred to a 1.5 ml Eppendorf tube in which it was spin down at 2500 rpm for 5 min at 4. After taking off the PBS, samples were stored at −20°C or directly used for further extraction. For sphingoid base analysis, samples were re-suspended in 150 uL extraction buffer (ethanol, water, diethyl ether, pyridine, and 4.2 N ammonium hydroxide (15:15:5:1:0.018, v/v)). A mixture of internal standards (0.04 nmol of C17 sphingosine, 0.04 nmol of C17 sphinganine, 0.4 nmol of C17 sphingosine-1-phosphate, 0.4 nmol of C17 sphinganine-1-phosphate) was added. The samples were vigorously vortexed using a Cell Disruptor Homogenizer(Disruptor Genie, Scientific Industries) for 10 min at 4°C and incubated on ice for 20 min. Cell debris were pelleted by centrifugation at 14,000 rpm for 2 min at 4°C, and the supernatant was collected. The extraction was repeated once more without ice incubation. The supernatants were combined and dried under vacuum in a CentriVap (Labconco, Kansas City, MO). The samples were re-suspended in a mixture of solvents composed of 70 µl of borate buffer (200 mM boric acid pH 8.8, 10 mM tris(2-carboxyethyl)-phosphine, 10 mM ascorbic acid and 33.7 µM $^{15}N^{13}C$-valine), and 10 µl of formic acid solution (0.1% aqueous solution), derivatized by reacting with 20 µl 6-aminoquinolyl-N-hydroxysuccinimidyl carbamate (AQC) solution (2.85 mg/ml in acetonitrile) for 15 min at 55°C. After overnight incubation at 24°C, samples were analyzed by LC-MS/MS in an Accela HPLC system (ThermoFisher Scientific, Waltham, USA) coupled to a TSQ Vantage (ThermoFisher Scientific, Waltham, USA). MRM-MS was used to identify and quantify sphingoid bases. The amounts of sphingolipids were normalized with respect to the amount of C17 internal standards and cell numbers.

For ceramide and phospholipid analysis, samples were prepared following the MTBE protocol (*Matyash et al., 2008*). Briefly, cells were re-suspended in 100 µL of water and transferred into a 2 ml Eppendorf tube. 360 µl of MeOH and a mixture of internal standards (0.4 nmol of DLPC, 1 nmol of PE31:1, 1 nmol of PI31:1, 3.3 nmol of PS31:1, 2.500 nmol of C12 sphingomyelin, 0.5 nmol of C17 ceramide and 0.1 nmol of C8 glucosylceramide) was added. Samples were vortexed, following the addition of 1.2 ml of MTBE. The samples were vigorously vortexed at maximum speed for 10 min at 4°C and incubated for 1 hr at room temperature on a shaker. Phase separation was induced by addition of 200 µL MS-grade water and incubation for 10 min. Samples were centrifuged at 1000 g for 10 min. The upper phase was transferred into a 13 mm glass tube and the lower phase was re-extracted with 400 µl of a MTBE/MeOH/H2O mixture (10:3:1.5, v/v). The extraction was repeated one more time. The combined upper phase was separated into three equal aliquots before drying under nitrogen flow. One aliquot was treated by alkaline hydrolysis to enrich for sphingolipids, one was used for glycerophospholipid analysis and the third was kept as a backup. To deacylate glycerophospholipids, the sample was re-suspended in 1 ml freshly prepared monomethylamine reagent (methylamine/H2O/n-butanol/methanol at 5:3:1:4 (v/v)) and incubated at 53 for 1 hr in a water bath before dried by nitrogen flow. The excess salts were removed by extracting samples in 300 µL water-saturated n-butanol solution and 150 µL MS-grade water (*Clarke and Dawson, 1981*). The organic phase was collected, and the extraction was repeated with 300 µL water-saturated n-butanol. The combined organic phase was dried by nitrogen flow.

Identification and quantification of phospholipid and sphingolipid molecular species were performed using multiple reaction monitoring with a TSQ Vantage Triple Stage Quadrupole Mass Spectrometer (Thermo Scientific) equipped with a robotic nanoflow ion source, Nanomate HD (Advion Biosciences). Each individual ion dissociation pathway was optimized with regard to collision energy. Lipid concentrations were calculated with respect to the corresponding internal standards and were presented as percentage of all lipid signals(*Guan et al., 2013*).

## Live-cell imaging

Subcellular localization experiments were performed on a Leica SP8 confocal microscope using a 63 x oil immersion objective. A 405 nm and 532 nm laser were used with appropriate filter settings during image acquisition. Photoactivation experiments were performed using a Nikon A1r microscope with 63 x oil immersion objective at 37 with 5% CO2. A 100 mW 402 nm laser was used for photo-releasing sphingosine, and a 488 nm laser was used for recording the time-lapse images. Specifically,

uncaging was performed within a circle in 3.0 μm diameter. Cells were irradiated for 4 s (2.2 μs/pixel) at 100% laser power in the region of interest (mitochondria in the case of Mito-So uncaging). Raw imaging data were analyzed by measuring the mean intensity of laser-illuminated cells.

## Image analysis

The fluorescence images were analyzed by Fiji software (*Schindelin et al., 2012*). The fluorescence staining images were presented in their original form. Time-lapse images were extracted by measuring the mean intensities of photo-stimulated individual cells during the acquisition. The single-cell traces were normalized to baseline of each cell before exporting to GraphPad Prism.

## Statistical analysis

Data represent at least the average of three independent experiments. Error bars represent standard error of the mean (SEM) as indicated. Statistical significance was calculated based on two-tailed unpaired student's *t*-test.

## Generation of mutant cells

Mutant cells were generated by the CRISPR/Cas9 system from Streptococcus pyogenes (*Ran et al., 2013*), using the HPRT co-targeting strategy (*Liao et al., 2015*) as previously described (*Harayama and Riezman, 2017*). Target sequences (listed below) were selected based on high specificity and efficacy scores predicted by the CRISPOR algorithm (*Haeussler et al., 2016*), and the corresponding pairs of oligo DNA were synthesized (Microsynth AG, Balgach, St. Gallen, Switzerland). Plasmids were constructed by assembling annealed oligo DNA pairs and plasmids in single tube reactions using FastDigest Bpi I (Thermo Fisher Scientific) and quick ligase (New England Biolabs, Ipswich, MA, USA) in quick ligase buffer. Three cycles of restriction at 37 and ligation at 25 were repeated (5 min for each step), followed by Bpi I restriction for one hour to remove empty vectors. pX330 plasmids (*Ran et al., 2013*) were used for assembly, except for HPRT guide RNA, for which pUC-U6-sg plasmid (*Harayama and Riezman, 2017*) was used. Plasmids were transformed into chemical competent STBL3 bacterial cells (Thermo Fisher Scientific, Waltham, MA, USA), sequence-verified by Sanger sequencing (Fasteris SA, Plan-les-Ouates, Geneva, Switzerland), and purified using GenElute Plasmid Miniprep kit (Sigma-Aldrich, St. Louis, MO, USA) followed by endotoxin removal by isothermal Triton X-114 extraction (*Ma et al., 2012*). Plasmids (98 ng total of plasmids for target(s) and 2 ng plasmid for HPRT) were reverse transfected into HeLa MZ cells using Lipofectamine 3000 (Thermo Fisher Scientific) in a 96 well plate, cell culture areas were scaled up before reaching over confluency, and selected with 6 μg/mL 6-thioguanine (Sigma-Aldrich) 5 days post-transfection. After one week of selection, the resistance against 6-thioguanine caused by the mutations in the co-targeted HPRT gene led to enrichment for mutations in the target genes (*Liao et al., 2015*). To evaluate mutation rates, individual loci were analyzed by PCR direct sequencing (using primers listed below) followed by TIDE (tracking indels by deconvolution) analysis (*Brinkman et al., 2014*). PCR reactions were peformed using ExTaq polymerase (TAKARA Clonthech, Otsu, Shiga, Japan).

- • Guide RNA sequences for genome editing
  - –SPHK1: CTGGTGCTGCTGAACCCGCG
  - –SPHK2: TGAGTGGGATGGCATCGTCA
  - –HPRT: GTAGCCCTCTGTGTGCTCAA
- • Primers for genome DNA analysis
  - –SPHK1: TATCCCTCACGAGGCCAGAA, TAGAGGAGCACTGACGGGAA
  - –SPHK2: AACCACGTGCTTCCCATGAT, GGGGTTGGGGAAAGAGACAG

## Chemical synthesis

$^1$H and $^{13}$C-NMR spectra were recorded on a Bruker AMX-400 MHz or 500 MHz spectrometer. Chemical shifts are given in ppm ($\delta$) using the NMR solvent as internal references and J values are reported in Hz. Splitting patterns are designated as follows: s, singlet; d, doublet; t, triplet; q, quartet; m, multiplet; b, broad. $^{13}$C-NMR spectra were broadband hydrogen decoupled. LC-MS spectra were recorded using a Thermo Electron Corporation HPLC with a Thermo Scientific Finnigan Surveyor MSQ Spectrometer System. High-resolution mass spectra were recorded on a QSTAR Pulsar

(QqTOF) mass spectrometer. Flash chromatography purification was carried out using silica gel 40–63 μm (200–400 mesh) from SiliCycle in solvent systems as described. Thin layer chromatography (TLC) was performed on aluminum-backed, pre-coated silica gel plates (Merck TLC silica gel 60 F254). Spots were detected by a UV lamp under 254 nm or 365 nm wavelength. Reverse phase HPLC purification was performed using an Agilent Technologies 1260 infinity HPLC equipped with a ZORBAX 300 SB-C18 column (9.4 × 250 mm). UV-Vis spectra were recorded using a JASCO V-650 spectrophotometer equipped with a stirrer and a temperature. Fluorescence measurements were performed with a FluoroMax-4 spectrofluorometer (Horiba Scientific) equipped with a stirrer and a temperature controller. (3-Aminopropyl)triphenylphosphonium bromide, compound 1 and subsequently methylated product were synthesized according to previously published procedures (*Kaur et al., 2015*; *Hagen et al., 2005*), respectively.

## 7-[(*tert*-butoxycarbonylmethyl)-methylamino]-4-methylcoumarin (2)

**Chemical structure 1.** Structure for 7-[(tert-butoxycarbonylmethyl)-methylamino]-4-methylcoumarin.
DOI: https://doi.org/10.7554/eLife.34555.022

To a suspension of sodium hydride (132 mg, 60% in mineral oil, 3.3 mmol, 1.1 equiv.) in 2 ml dry DMF at 0, a solution of 7-(*tert*-butoxycarbonylmethyl)−4-methylcoumarin (870 mg, 3.0 mmol) (*Hagen et al., 2005*) in 8 ml DMF was added via a syringe under inert atmosphere, followed by the addition of methyl iodide (600 μL, 9.7 mmol). The reaction mixture was stirred at 0 for 1.5 hr. The reaction mixture was quenched by 0.1 N HCl solution (40 mL) and extracted with EtOAc (40 mL). The organic phase was washed twice with brine, dried over Na2SO4, and concentrated under reduced pressure. The crude product was purified by flash chromatography (SiO2, EtOAc/cyclohexane = 1/5) to give compound 2 (800 mg, Yield: 88%) as a yellow solid. Rf = 0.40 (EtOAc/cHex = 1/4). $^1$H NMR (400 MHz, CDCl3) $\delta$ = 7.41 (d, J = 8.9 Hz, 1H), 6.59 (dd, J = 8.9, 2.6 Hz, 1H), 6.51 (d, J = 2.6 Hz, 1H), 6.00 (d, J = 1.1 Hz, 1H), 4.01 (s, 2H), 3.12 (s, 3H), 2.35 (d, J = 1.1 Hz, 3H), 1.44 (s, 9H). $^{13}$C NMR (101 MHz, CDCl3) $\delta$ = 169.09, 162.14, 155.71, 152.96, 152.01, 125.59, 110.72, 110.10, 109.04, 98.98, 82.45, 55.10, 39.99, 28.22, 18.62 ppm. HR-ESI-MS (pos.) C17H21NO4, [M + H]$^+$ calculated: 304.1543, [M + H]$^+$ found: 304.1532.

## 7-[(*tert*-butoxycarbonylmethyl)-methylamino]−4-(hydroxymethyl)coumarin (3)

**Chemical structure 2.** Strucutre for 7-[(tert-butoxycarbonylmethyl)-methylamino]-4-(hydroxymethyl)coumarin.
DOI: https://doi.org/10.7554/eLife.34555.023

To a solution of compound 2 (303 mg, 1.0 mmol) in xylene (15 mL), selenium dioxide (220 mg, 2.0 mmol) was added. After stirring at 145 for 20 hr under nitrogen atmosphere, the reaction mixture was filtered and concentrated *in vacuo*. The crude product was re-dissolved in 20 mL methanol, followed by addition of sodium borohydride (95 mg, 2.5 mmol). The reaction was stirred at room temperature for 1 hr before quenching with EtOAc (20 mL) and water (20 mL). The organic phase was washed twice with brine, dried over Na2SO4, and concentrated under reduced pressure. The crude product was purified by flash chromatography (SiO2, EtOAc/cyclohexane = 1/2) to give compound 4 (183 mg, Yield: 58%) as a light yellow solid. $^1$H NMR (400 MHz, CDCl3) $\delta$ = 7.26 (d, J = 9.0 Hz, 1H, overlapped with solvent peak), 6.52 (dd, J = 9.0, 2.6 Hz, 1H), 6.45 (d, J = 2.6 Hz, 1H), 6.29 (s, 1H), 4.75 (s, 2H), 3.99 (s, 2H), 3.09 (s, 3H), 1.43 (s, 9H) ppm. $^{13}$C NMR (101 MHz, CDCl3) $\delta$ = 169.19,

162.56, 155.51, 155.13, 151.80, 124.21, 108.96, 107.70, 106.35, 98.67, 82.47, 60.63, 54.83, 39.72, 28.08 ppm. HR-ESI-MS (pos.) C17H21NO5, [M + H]$^+$ calculated: 320.1493, [M + H]$^+$ found: 320.1484.

## 7- [(carboxymethyl)-methylamino]—4-(hydroxymethyl)coumarin (5)

**Chemical structure 3.** Structure for 7- [(carboxymethyl)-methylamino]-4-(hydroxymethyl)coumarin.
DOI: https://doi.org/10.7554/eLife.34555.024

Compound 3 (65 mg, 0.20 mmol) was slowly added to a mixture of TFA/CH2Cl2/H2O (75/25/1) at 0, and the reaction was stirred at room temperature for 3 hr. Small amount of toluene was added to the crude mixture, and the solvents were evaporated under reduced pressure. The crude mixture was lyophilized to give a yellow solid. The product was direct used for next step without further purification. $^1$H NMR (400 MHz, Methanol-d5) δ = 7.46 (d, J = 9.0 Hz, 1H), 6.71 (dd, J = 9.0, 2.6 Hz, 1H), 6.56 (d, J = 2.6 Hz, 1H), 6.27 (t, J = 1.31 Hz, 1H), 4.79 (d, J = 1.5 Hz, 2H), 4.23 (s, 2H), 3.14 (s, 3H). $^{13}$C NMR (101 MHz, Methanol-d4) δ = 173.56, 164.79, 158.59, 156.79, 153.80, 125.64, 110.56, 108.65, 106.01, 99.24, 60.79, 54.30, 39.71 ppm. ESI-MS (pos.) C13H13NO5, [M + H]$^+$ calculated: 264.09, [M + H]$^+$ found: 264.13.

## TPP-Cou-OH (4)

**Chemical structure 4.** Structure for TPP-Cou-OH.
DOI: https://doi.org/10.7554/eLife.34555.025

To a solution of compound 5 obtained from the last step and (3-Aminopropyl)triphenylphosphonium bromide (80 mg, 0.20 mmol) in 10 ml of dry DCM, DIPEA (87 μL, 0.50 mmol) and HBTU (80 mg, 0.21 mmol) were added. The reaction mixture was stirred at room temperature for overnight under nitrogen atmosphere. The solvents were evaporated under reduced pressure. The crude product was purified by flash chromatography (SiO2, 10% DCM in MeOH) to give 4 (33 mg, Yield: 25% over two steps) as a yellow solid. $^1$H NMR (500 MHz, CDCl3) δ = 8.83 (s, 1H), 7.78 (m, 3H), 7.58–7.67 (m, 12H), 7.00 (d, 1H, J = 9.0 Hz), 6.60 (d, 1H, J = 9.0 Hz), 6.39 (d, 1H, J = 2.0 Hz), 6.07 (s, 1H), 4.50 (s, 2H), 4.09 (s, 2H), 3.50 (m, 2H), 3.32 (m, 2H), 3.13 (s, 3H), 1.89 (m, 2H) ppm. $^{13}$C NMR (126 MHz, CDCl3, DEPT) δ = 135,25, 135.23, 133.23, 133.15, 130.56, 130.47, 123.84, 109.22, 105.72, 98.32, 60.00, 56.00, 39.95, 39.00, 38.86, 22.35, 22.33, 20.40, 19.98 ppm. HR-ESI-MS (pos.) C34H34N2O4P, [M]$^+$ calculated: 565.2251, [M]$^+$ found: 565.2255.

Mito-So

**Chemical structure 5.** Structure for Mito-So.
DOI: https://doi.org/10.7554/eLife.34555.026

To a solution of TPP-Cou-OH (4) (26 mg, 0.04 nmol) in 4 mL of dry DMF, Bis-(4-nitrophenyl)carbonate (12 mg, 0.04 mmol) and DIPEA (20 μL, 0.11 mmol) was added. The reaction mixture was stirred at room temperature in dark. After 3 hr, sphingosine (12 mg, 0.04 mmol) in 1 mL of DMF was added and the reaction was stirred at 60 for 3 hr in dark. The crude product was directly purified by flash chromatography (SiO2, MeOH/DCM = 1/10) to afford Mito-So (15 mg, Yield: 39%) as a yellow solid. The product was further purified by reverse phase HPLC. Semi-preparative HPLC runs were carried out with a gradient from 5% to 95% acetonitrile/water system (0.1% TFA) for 20 min and a flow of 1 mL/min, monitored by a PDA detector at 254 nm and 360 nm. The fraction was collected and lyophilized to afford 4.85 mg of the product. [1]H NMR (500 MHz, DMSO-d6) $\delta$ = 8.15 (t, 1H, J = 5.9 Hz), 7.90 (m, 3H), 7.74–7.78 (m, 12H), 7.42 (d, 1H, J = 9.0 Hz), 7.15 (d, 1H, J = 9.0 Hz), 7.66 (dd, 1H, J = 9.0, 2.5 Hz), 6.53 (d, 1H, J = 2.5 Hz), 6.06 (s, 1H), 5.55 (m, 1H), 5.43 (m, 1H), 4.05 (s, 2H), 3.91 (t, 1H, J = 7.0 Hz), 3.51–3.47 (m, 7H, overlapped with water peak), 3.26 (q, 2H, J = 6.3 Hz), 3.08 (s, 3H), 1.94 (q, 2H, J = 6.8 Hz), 1.67 (m, 2H), 1.17–1.26 (m, 22H), 0.85 (t, 3H, J = 6.9 Hz) ppm. [13]C NMR (126 MHz, DMSO-d6, DEPT) $\delta$ = 135.47, 135.44, 134.01, 133.93, 131.69, 131.60, 130.78, 130.68, 125.26, 109.58, 105.54, 98.20, 71.85, 70.25, 61.01, 58.31, 55.49, 39.18, 39.02, 29.55, 29.53, 29.51, 29.49, 29.45, 29.23, 29.18, 29.03, 22.73, 22.57, 18.97, 18.56, 14.44 ppm. HR-ESI-MS (pos.) C53H69N3O7P, [M]+ calculated: 890.4868, [M]+ found: 890.4860.

Mito-Sa

**Chemical structure 6.** Structure for Mito-Sa.
DOI: https://doi.org/10.7554/eLife.34555.027

Mito-Sa was synthesized following the same procedures as of Mito-So and 19 mg of Mito-Sa was obtained (Yield: 50%). Semi-preparative HPLC was used to further purify the product to afford 4 mg of Mito-Sa. [1]H NMR (500 MHz, DMSO-d$_6$) $\delta$ = 8.12 (t, 1H, J = 5.8 Hz), 7.89 (m, 3H), 7.73–7.77 (m, 12H), 7.42 (d, 1H, J = 9.0 Hz), 7.19 (d, 1H, J = 9.0 Hz), 6.65 (dd, 1H, J = 9.0, 2.5 Hz), 6.52 (d, 1H, J = 2.5 Hz), 6.05 (s, 1H), 5.14–5.23 (m, 2H), 4.04 (s, 2H), 3.46–3.63 (m, 7H, overlapped with water peak), 3.25 (q, 2H, J = 6.40 Hz), 3.07 (s, 3H), 1.67 (m, 2H), 1.43 (m, 2H), 1.17–1.28 (m, 25H), 0.84 (t,

3H, J = 7.0 Hz) ppm. $^{13}$C NMR (126 MHz, DMSO-$d_6$, DEPT) $\delta$ = 134.92, 134.90, 133.47, 133.39, 130.24, 130.14, 124.79, 109.06, 104.98, 97.66, 69.87, 60.67, 60.53, 57.73, 54.97, 38.63, 38.48, 33.31, 31.22, 29.09, 29.04, 28.99, 28.94, 28.63, 25.10, 22.19, 22.03, 18.44, 18.02, 13.90 ppm. HR-ESI-MS (pos.) $C_{53}H_{71}N_3O_7P$, [M]$^+$ calculated: 892.5024, [M]$^+$ found: 892.5021.

## Acknowledgements

We thank Roman Lagoutte for helpful discussion, Isabelle Riezman and Thomas Hannich for technical support in lipidomic experiments, and all other members of the Riezman lab for insightful comments. We thank the Bioimaging Center (University of Geneva) and Bioimaging Core Facility (University of Geneva) for technical assistance. TH was supported by the Japan Society for the Promotion of Science (JSPS) Postdoctoral Fellowships for Research Abroad. This study was supported by Sinergia, the Swiss National Science Foundation (CRSII3-154405) and the NCCR Chemical Biology funded by the Swiss National Science Foundation (51NF40-160589).

## Additional information

### Funding

| Funder | Grant reference number | Author |
| --- | --- | --- |
| Schweizerischer Nationalfonds zur Förderung der Wissenschaftlichen Forschung | CRSII3-154405 | Howard Riezman |
| National Centre for Competence in Research in Chemical Biology | 51NF40-160589 | Nicolas Winssinger Howard Riezman |
| Japan Society for the Promotion of Science | | Takeshi Harayama |

The funders had no role in study design, data collection and interpretation, or the decision to submit the work for publication.

### Author contributions

Suihan Feng, Conceptualization, Data curation, Formal analysis, Validation, Investigation, Visualization, Methodology, Writing—original draft, Project administration; Takeshi Harayama, Conceptualization, Data curation, Formal analysis, Funding acquisition, Validation, Investigation, Visualization, Methodology, Writing—review and editing; Sylvie Montessuit, Investigation, Methodology, Writing—review and editing; Fabrice PA David, Data curation, Writing—review and editing; Nicolas Winssinger, Resources, Funding acquisition, Writing—review and editing; Jean-Claude Martinou, Conceptualization, Supervision, Funding acquisition, Writing—review and editing; Howard Riezman, Conceptualization, Supervision, Funding acquisition, Writing—original draft, Project administration, Writing—review and editing

### Author ORCIDs

Suihan Feng (iD) http://orcid.org/0000-0002-9205-0050
Howard Riezman (iD) http://orcid.org/0000-0003-4680-9422

### Ethics

Animal experimentation: All experimental procedures were performed according to guidelines provided by the Animal Welfare Act and Animal welfare ordinance, the Rectors' Conference of the Swiss Universities (CRUS) policy and the Swiss Academy of Medical Sciences / Swiss Academy of Sciences' Ethical Principles and Guidelines for Experiments on Animals, and were approved by the Geneva Cantonal Veterinary Authority (authorization number: 28038/GE86/16).

### Decision letter and Author response

Decision letter https://doi.org/10.7554/eLife.34555.030

Author response https://doi.org/10.7554/eLife.34555.031

---

## Additional files

**Supplementary files**
• Transparent reporting form
DOI: https://doi.org/10.7554/eLife.34555.028

---

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
