## [Decision Letter]

[Editors’ note: a previous version of this study was rejected after peer review, but the authors submitted for reconsideration. The first decision letter after peer review is shown below.]

Thank you for submitting your work entitled "Mitochondria-specific photoactivation to monitor local sphingosine metabolism and function" for consideration by *eLife*. Your article has been reviewed by three peer reviewers, and the evaluation has been overseen by Suzanne Pfeffer as the Reviewing Editor and a Senior Editor. The reviewers have opted to remain anonymous.

The decision was reached after consultation between the reviewers. Unfortunately, based on these discussions and the individual reviews below, we regret to inform you that your work will not be considered further for publication in *eLife*. The reviewers felt that the study had for the most part, been carried out with great care and rigor. However, they felt that significant additional work would be needed to show that the uncaged sphingosine phosphorylation actually takes place in mitochondria. Given their additional concerns, it was not possible to come to a more favorable disposition. Nevertheless, we hope you will find the comments constructive in revising the story for another journal.

Reviewer #1:

This paper describes the synthesis and use of a novel, mitochondria targeted and caged derivative of sphingosine as a new approach intended for the investigation of compartment-specific sphingosine metabolism in the mitochondria. However, as implemented here, this approach does lacks confirmation that phosphorylation of sphingosine to sphingosine-1-phosphate (S1P) by sphingosine kinase 1 (SphK1) occurs in the mitochondria. Based on the finding that SphK1 is required and that SphK2 is known to be localized to the mitochondria, it is likely that sphingosine that is very rapidly released by uncaging in the mitochondria is rapidly transferred out of the mitochondria into other cellular compartments that contain sphingosine kinase enzyme activity as well as other sphingolipid metabolic enzymes. There are several major concerns, including evidence that levels of S1P are actually increased in the mitochondria, that mitochondrial functions are affected, and lack of evidence that SphK1 is present in mitochondria.

Several groups have reported that S1P has various physiological functions in the mitochondria. Although it is shown here that un-caging of mitochondrial targeted sphingosine is rapidly converted to S1P in cells, a reaction the authors consider takes place in the mitochondria, evidence is lacking to show that S1P generated in the mitochondria has a function in there. It is also important to substantiate that the increase in cellular S1P actually takes place in the mitochondria.

It has previously been shown that SphK2 is the main sphingosine kinase isoenzyme present in mitochondria. There is no published evidence to date that shows that SphK1 is also in mitochondria. It is necessary to determine whether SphK1 is a mitochondrial resident protein or not. If not, it is difficult to understand how conversion of sphingosine to S1P occurs in the mitochondria.

Subsection “Uncaged sphingosine and sphinganine are rapidly phosphorylated”, fourth paragraph: It is misleading to state that SphK1 and SphK2 carry out redundant functions. This was not shown by Mizugishi et al., 2005. Both SphK1 and SphK2 produce S1P, but their presence in different cellular locations suggests that they have compartment specific functions. Although several reports have shown that circulating S1P levels are increased in SphK1 knockout mice, S1P is produced by a compensatory increase in SphK2. This does not indicate that SphK1 and SphK2 have redundant functions but may have some compensatory functions.

Subsection “Metabolism of locally uncaged sphingosine”, first paragraph: This sentence is incorrect. "local lipid metabolism" was not monitored at all in this study. Total cellular levels of some sphingolipid metabolites, not local levels, were measured at various times. This was probably at some time after UV uncaging, although this was not stated in the figure legend. It is important to note that whole cell turnover of sphingosine and S1P is a complex process that involves synthesis, metabolism, and degradation. This approach does not reveal anything about rate of conversion of sphingosine to S1P or sphingosine kinase activity since the synthesis and degradation takes place in several subcellular compartments. Therefore, the half-life data in Figure 5 cannot be interpreted as showing that SphK1 is more important than SphK2.

Figure 5: The figure title is not correct – only total cellular levels of sphingosine and S1P were measured, not mitochondrial levels. The figure legend does not explain if time (min) is after loading and then uncaging. Why are the S1P levels at 0 min in the SphK1/SphK2 double knockout cells only about 10% of those in the wild type cells? There should be no S1P in cells without SphK1 and SphK2 as S1P can only be produced by SphKs. Expression levels of SphKs should be determined and if present, cells should be characterized as partial deletions, not knockouts. The data on ceramides in panels C and F are meaningless as presented. They are expressed as mol% of total measured cellular lipids, not just sphingolipids, that make up around 99%. These cannot be compared with sphingosine and S1P levels expressed as pmol/million cells. Moreover, expected changes in ceramide levels produced from uncaged sphingosine would be infinitesimal on this mol% basis. All lipid quantitation data should be expressed in molar units per cell, or mg protein, etc.

Discussion, third paragraph: As noted by the authors, examination of flux from mitochondria-localized sphingosine into ceramide requires the use of mito-caged, isotope labeled sphingosine. The data in this paper do not address this flux at all and the interpretation and discussion should be revised to clearly indicate this drawback of this study. Furthermore, the statement in the last paragraph of the Discussion should be deleted since this study did not "prove" anything about the importance of subcellular localization in lipid signaling and metabolism or even about sphingosine/S1P signaling and metabolism.

Subsection “Lipid extraction and quantification”, first paragraph: It is not clear why this "non-traditional" method was used for extraction of lipids as the references cited either do not contain any relevant methods or involve the determination of sphingoid bases (reference to dos Santos et al. is missing). Internal standards were used, but there is no data on recovery or linearity of assays that should be provided. Furthermore, the 2008 method cited in the second paragraph of the aforementioned subsection for extraction of ceramides has not been validated for ceramides or sphingoid bases by the original authors or by the present authors. Moreover, it is not proper to present quantitative data as percentage of all lipid signals. This does not minimize technical variance, it covers them up and hides changes in minor but important lipids.

*Reviewer #2:*

This manuscript from the Riezman lab characterized a novel mitochondrial targeted caged sphingosine. The authors show that the probe is organelle-specific and that efficiently released sphingosine which was then metabolized into S1P. The authors also show that mito-caged Sphingosine did not induce intracellular calcium levels.

The work is well done the manuscript is written in a clear and lucid manner.

Overall, the impact of the work is limited. The key issue is that the findings are incremental and largely negative or confirmatory. The authors did not show what is the function of mito-targeted sphingosine. In my opinion, this is needed to complete the story. Overall, the work does add to the literature by describing a unique tool – but I feel that this is of interest to a specialized audience and at this point, the work is not of general interest to the larger community.

1) Figure 3 – conversion (phosphorylation) rate of mito-targeted sphingosine is around 25% whereas sphinganine is ~3%. The authors do not discuss this. Is this due to increased metabolism of the latter lipid to Ceramide species?

2) The concept that sphk isoenzymes are localized in the mitochondria is not fully supported by data – other than the few citations that the authors mention (i.e. Strub et al., 2011). The primary data regarding this issue is not unequivocal. The authors data can also be explained by diffusion of released sphingosine and metabolism in other compartments. Just because of two types of caged sphingosines show different effects on calcium, the authors cannot say for sure that diffusion does not occur post-uncaging.

3) Even though the authors address the very low sphingosine levels in their control KD cells (Figure 4), the cells have 10-fold less basal levels of this lipid. This raises the question of why this is. Is this due to increased sphingosine kinase expression or ceramics synthase expression? Or is de novo sphingolipid synthesis pathway suppressed? This could have an effect on cellular response to sphingolipid mediators so should be addressed.

4) In Figure 5, the authors only showed total ceramide levels. Upon uncaging and release of sphingosine, only certain ceramide species are likely altered. Thus, the authors should show individual ceramide species which would allow them to make more definitive statements about ceramide synthase steps.

*Reviewer #3:*

In this manuscript, Feng et al. develop a method to release sphingosine specifically in the mitochondria and follow its fate by mass spectrometry. Their data suggest that sphingosine kinases are active within mitochondria, and that the mitochondrial pool of S1P has distinct functions from other subcellular pools. The tool developed is innovative and useful, the results underscore the importance of subcellular control of sphingolipid distribution, and the paper is very clearly written.

My comments come with the caveat that I am not a specialist in lipid chemistry. My only two concerns which might require new experiments are:

1) Figure 2. This is the figure that demonstrates that Mito-So is imported by functional mitochondria, a key point, but all the stains are diffuse. The authors don't provide detail on which MitoTracker they use for the stain (this should be added to the Materials and methods), but presumably they use one that stains functional mitochondria, which is why upon the addition of CCCP both the Mito-So and MitoTracker stains lose definition. It would be nice to add a stain of total mitochondria to show that the cells aren't just dead and sticky.

Figure 4. Could the authors please discuss why they find little reduction in basal S1P in Sphk1/2 KO cells compared controls? Is the signal dominated by S1P from the FBS in the cell culture medium, are they at the limit of detection of their assay, or is something else going on?

---

## [Author Response]

[Editors’ note: the author responses to the first round of peer review follow.]

Reviewer #1:This paper describes the synthesis and use of a novel, mitochondria targeted and caged derivative of sphingosine as a new approach intended for the investigation of compartment-specific sphingosine metabolism in the mitochondria. However, as implemented here, this approach does lacks confirmation that phosphorylation of sphingosine to sphingosine-1-phosphate (S1P) by sphingosine kinase 1 (SphK1) occurs in the mitochondria. Based on the finding that SphK1 is required and that SphK2 is known to be localized to the mitochondria, it is likely that sphingosine that is very rapidly released by uncaging in the mitochondria is rapidly transferred out of the mitochondria into other cellular compartments that contain sphingosine kinase enzyme activity as well as other sphingolipid metabolic enzymes. There are several major concerns, including evidence that levels of S1P are actually increased in the mitochondria, that mitochondrial functions are affected, and lack of evidence that SphK1 is present in mitochondria.

Even though we do not find this explanation likely, we accept this criticism and see the need to provide further evidence for S1P formation in mitochondria. To confirm that the phosphorylation of photoreleased sphingosine can occur inside mitochondria, we performed the uncaging assay using purified mitochondria from mouse liver, and observed a significant increase in S1P after uncaging. Together with control experiments showing that if S1P had been formed outside the mitochondria it would not be detected in our assay, our results provide clear evidence that sphingosine can be phosphorylated inside mitochondria. In addition, our studies on metabolism and signalling do not support a model that the mitochondrial sphingosine can easily diffuse in the cell. The new data in Figure 7 clearly show that sphingosine generates a different metabolic pattern depending on the location of the subcellular source. More importantly, we did not detect any calcium spike after photoreleasing mitochondrial sphingosine, in contrast to the calcium spikes after globally uncaging sphingosine. Taken together, our data show that sphingosine metabolism and signalling are highly dependent on its the subcellular localization, which is really the main point of the study.It is technically very difficult to quantify the movement of lipids intracellularly, but our data show, for the first time, that sphingosine or S1P can exit mitochondria and provide a framework to study the kinetics of this process.

Several groups have reported that S1P has various physiological functions in the mitochondria. Although it is shown here that un-caging of mitochondrial targeted sphingosine is rapidly converted to S1P in cells, a reaction the authors consider takes place in the mitochondria, evidence is lacking to show that S1P generated in the mitochondria has a function in there. It is also important to substantiate that the increase in cellular S1P actually takes place in the mitochondria.

To investigate in this issue, we performed an uncaging assay in purified mitochondria from mouse liver, and observed a significant increase in S1P after uncaging.

It has previously been shown that SphK2 is the main sphingosine kinase isoenzyme present in mitochondria. There is no published evidence to date that shows that SphK1 is also in mitochondria. It is necessary to determine whether SphK1 is a mitochondrial resident protein or not. If not, it is difficult to understand how conversion of sphingosine to S1P occurs in the mitochondria.

We can understand the reviewer’s point here, but this is not the intended focus of our manuscript. Our goal was not to determine the subcellular localization of sphingosine kinases. Our new data showing S1P formation in mitochondria shows that there must be sphingosine kinase activity in the organelle. This is enough for us. We chose not to localize the sphingosine kinases for two reasons, one, we feel that it distracts from the main point of the manuscript, two, it is quite possible that the localization of the two sphingosine kinases depends on the cell type. Therefore, without a physiological readout of S1P function in the mitochondria, in our opinion, the localization experiments have little added value. To make our point that S1P formation is enzymatic, we have only included the double knockout of both kinases to show this and have removed the single mutants from the new version of the manuscript.

Subsection “Uncaged sphingosine and sphinganine are rapidly phosphorylated”, fourth paragraph: It is misleading to state that SphK1 and SphK2 carry out redundant functions. This was not shown by Mizugishi et al., 2005. Both SphK1 and SphK2 produce S1P, but their presence in different cellular locations suggests that they have compartment specific functions. Although several reports have shown that circulating S1P levels are increased in SphK1 knockout mice, S1P is produced by a compensatory increase in SphK2. This does not indicate that SphK1 and SphK2 have redundant functions but may have some compensatory functions.

This was really not the point of the manuscript and now with the single mutants removed this comment is no longer relevant. Our meaning of redundant in this case was they both phosphorylate sphingosine and as far as we know their substrate specificity is the same.

Subsection “Metabolism of locally uncaged sphingosine”, first paragraph: This sentence is incorrect. "local lipid metabolism" was not monitored at all in this study. Total cellular levels of some sphingolipid metabolites, not local levels, were measured at various times. This was probably at some time after UV uncaging, although this was not stated in the figure legend. It is important to note that whole cell turnover of sphingosine and S1P is a complex process that involves synthesis, metabolism, and degradation. This approach does not reveal anything about rate of conversion of sphingosine to S1P or sphingosine kinase activity since the synthesis and degradation takes place in several subcellular compartments. Therefore, the half-life data in Figure 5 cannot be interpreted as showing that SphK1 is more important than SphK2.

We use “Metabolism of locally uncaged sphingosine” as the title of a subsection. Since we performed uncaging in a specific subcellular location (mitochondria), and that most of the detected sphingosine comes after uncaging, it is correct to state it in this way. While sphingolipid homeostasis is a very complex process, our data demonstrate a role of subcellular localization in sphingolipid metabolism. We did not suggest that SphK1 is more important than SphK2. Some of the sphingosine phosphorylation may take place inside the mitochondria and some outside. In fact, this could be expected. We would like to point out that there are no data showing an export of sphingosine or sphingosine 1P from mitochondria, but this is clearly now the case with in our study.

Figure 5: The figure title is not correct – only total cellular levels of sphingosine and S1P were measured, not mitochondrial levels. The figure legend does not explain if time (min) is after loading and then uncaging. Why are the S1P levels at 0 min in the SphK1/SphK2 double knockout cells only about 10% of those in the wild type cells? There should be no S1P in cells without SphK1 and SphK2 as S1P can only be produced by SphKs. Expression levels of SphKs should be determined and if present, cells should be characterized as partial deletions, not knockouts. The data on ceramides in panels C and F are meaningless as presented. They are expressed as mol% of total measured cellular lipids, not just sphingolipids, that make up around 99%. These cannot be compared with sphingosine and S1P levels expressed as pmol/million cells. Moreover, expected changes in ceramide levels produced from uncaged sphingosine would be infinitesimal on this mol% basis. All lipid quantitation data should be expressed in molar units per cell, or mg protein, etc.

In the new manuscript, we now changed the title of Figure 6 to “Real-time metabolism of mito-caged sphingosine and globally-caged sphingosine in living cells”. To help readers to understand the experiments, we added Figure 6 to illustrate the design of the experiments, which also clarifies the point about the timing of the experiment. The S1P level in SphK dKO cells reflects the technical limit of the detection. Nevertheless, we have decided to remove the data on S1P and ceramides in this figure because the derivation of the half-life of S1P encompasses two events, generation of S1P from sphingosine as well as S1P metabolism and because we have introduced a new experiment that clearly shows the flux from localized sphingosine into ceramide using heavy-isotope labelled sphingosine. For information, the knockout efficiency of the kinases in the dKO cells is close to complete (> 95% ) as demonstrated in Figure 4—figure supplement 1. Our comments on the use of mole% are below.

Discussion, third paragraph: As noted by the authors, examination of flux from mitochondria-localized sphingosine into ceramide requires the use of mito-caged, isotope labeled sphingosine. The data in this paper do not address this flux at all and the interpretation and discussion should be revised to clearly indicate this drawback of this study. Furthermore, the statement in the last paragraph of the Discussion should be deleted since this study did not "prove" anything about the importance of subcellular localization in lipid signaling and metabolism or even about sphingosine/S1P signaling and metabolism.

We have added in an experiment examining the flux using heavy-isotope labelled sphingosine, so this comment is no longer relevant. We clearly show that mitochondrial sphingosine can find its way into ceramide and sphingomyelin.

Subsection “Lipid extraction and quantification”, first paragraph: It is not clear why this "non-traditional" method was used for extraction of lipids as the references cited either do not contain any relevant methods or involve the determination of sphingoid bases (reference to dos Santos et al. is missing). Internal standards were used, but there is no data on recovery or linearity of assays that should be provided. Furthermore, the 2008 method cited in the second paragraph of the aforementioned subsection for extraction of ceramides has not been validated for ceramides or sphingoid bases by the original authors or by the present authors. Moreover, it is not proper to present quantitative data as percentage of all lipid signals. This does not minimize technical variance, it covers them up and hides changes in minor but important lipids.

Our extraction method for sphingoide bases has been used and published before (Höglinger et al., 2015; Guri et al., 2017 (just published, referenced in the new version of the manuscript). There is a good description of the method in the STAR methods of the latter article. It is an excellent method for sphingoid bases and amino acids. We are preparing a manuscript for an analytical chemistry journal on the standard curves etc., but the description is a manuscript in itself and would be much too long. The extraction methods for other lipids including ceramides and sphingolipids is commonly used (Matyash et al., 2008). We simply do not agree about the use of percentage of all lipid signals. This does not cover up technical variance, but provides a method for normalization. The method reaches limitations when one has strong variations in a major lipid that then makes other lipid species appear to vary in the opposite direction, but this is not the case here. Sphingoid bases are minor lipids, as mentioned above and are always reported in either relative or in pmoles related to number of cells, not mol%.

Reviewer #2:[…] Overall, the impact of the work is limited. The key issue is that the findings are incremental and largely negative or confirmatory. The authors did not show what is the function of mito-targeted sphingosine. In my opinion, this is needed to complete the story. Overall, the work does add to the literature by describing a unique tool – but I feel that this is of interest to a specialized audience and at this point, the work is not of general interest to the larger community.

We have increased the importance of the findings by adding in experiments on the flux of mitochondrially released sphingosine and proving that phosphorylation can take place in mitochondria. This shows, for the first time, that sphingosine can be exported outside of the mitochondria. The only way this was possible was with the novel and unique tool that we have created and we believe that this, as a first in kind tool, is of great interest. We feel that this approach will be the one that will permit us and other to answer important questions on the role of lipid localization on metabolism and signalling. This manuscript clearly gives an example of this.

1) Figure 3 – conversion (phosphorylation) rate of mito-targeted sphingosine is around 25% whereas sphinganine is ~3%. The authors do not discuss this. Is this due to increased metabolism of the latter lipid to Ceramide species?

We now added a sentence in the manuscript. It is possible that sphingosine is better recognized by sphingosine kinases as a substrate, and/or that Mito-Sa is more efficiently accumulated in the cells. Most of the sphingoid base phosphates in cells are sphingosine 1P, so we prefer the latter. This is logical because sphinganine is probably mainly used for biosynthesis and sphingosine is derived from degradation, a process that is more often linked to S1P action.

2) The concept that sphk isoenzymes are localized in the mitochondria is not fully supported by data – other than the few citations that the authors mention (i.e. Strub et al., 2011). The primary data regarding this issue is not unequivocal. The authors data can also be explained by diffusion of released sphingosine and metabolism in other compartments. Just because of two types of caged sphingosines show different effects on calcium, the authors cannot say for sure that diffusion does not occur post-uncaging.

To further support the phosphorylation in mitochondria we performed controlled experiments with isolated mitochondria and showed that they can phosphorylation caged Mito-sphingosine.

*3) Even though the authors address the very low sphingosine levels in their control KD cells (Figure 4), the cells have 10-fold less basal levels of this lipid. This raises the question of why this is. Is this due to increased sphingosine kinase expression or ceramics synthase expression? Or is* de novo *sphingolipid synthesis pathway suppressed? This could have an effect on cellular response to sphingolipid mediators so should be addressed.*

We detected 4.2 times more sphingosine signals in the SphK dKO cells, probably because the S1P pathway is blocked leading to its accumulation. For the genetic experiments we used the HeLa MZ (from Marino Zerial) cells to generate the SphK KO cells because they are more efficiently modified than other HeLa lines that we have tried. The different basal levels of sphingosine are most likely due to the intrinsic difference between cell lines.

4) In Figure 5, the authors only showed total ceramide levels. Upon uncaging and release of sphingosine, only certain ceramide species are likely altered. Thus, the authors should show individual ceramide species which would allow them to make more definitive statements about ceramide synthase steps.

We now showed the major ceramide levels in Figure 7—figure supplement 1.

Reviewer #3:[…] My comments come with the caveat that I am not a specialist in lipid chemistry. My only two concerns which might require new experiments are:1) Figure 2. This is the figure that demonstrates that Mito-So is imported by functional mitochondria, a key point, but all the stains are diffuse. The authors don't provide detail on which MitoTracker they use for the stain (this should be added to the Materials and methods), but presumably they use one that stains functional mitochondria, which is why upon the addition of CCCP both the Mito-So and MitoTracker stains lose definition. It would be nice to add a stain of total mitochondria to show that the cells aren't just dead and sticky.

We used the MitoTracker Orange CMTMRox in our studies. We now added the details in the Materials and methods section. CCCP is commonly used chemical that disrupt the mitochondria potential but does not have acute cell toxicity, and thus we did not provide the images of mitochondria staining. The effects of CCCP on mitochondrial structures have been published by others previously and the mitochondrial staining never looks diffuse like we see with our probe under the influence of CCCP.

Figure 4. Could the authors please discuss why they find little reduction in basal S1P in Sphk1/2 KO cells compared controls? Is the signal dominated by S1P from the FBS in the cell culture medium, are they at the limit of detection of their assay, or is something else going on?

S1P level in mammalian cells is very low. The small S1P signal in SphK dKO cells reflects the technical boundary of our detection. It is unlikely that the S1P signal comes from the cell culture medium because of the extensive washing steps prior to lipid extraction and also because S1P is very soluble in aqueous buffer. There might be alternative pathways for S1P production. For instance, one could imagine a deacylation of ceramide 1P. So it would not be entirely unreasonable to still find some S1P in the double KO.